# Soil contamination in nearby natural areas mirrors that in urban greenspaces worldwide

Yu-Rong Liu [1,2] ✉, Marcel G. A. van der Heijden [3,4], Judith Riedo [4], Carlos Sanz-Lazaro [5,6], David J. Eldridge [7], Felipe Bastida [8], Eduardo Moreno-Jiménez [9,10,11], Xin-Quan Zhou[2], Hang-Wei Hu [12], Ji-Zheng He [12], José L. Moreno [8], Sebastian Abades [13], Fernando Alfaro [13,14], Adebola R. Bamigboye[15], Miguel Berdugo [16], José L. Blanco-Pastor[17], Asunción de los Ríos [18], Jorge Duran[19,20], Tine Grebenc [21], Javier G. Illán[22], Thulani P. Makhalanyane [23], Marco A. Molina-Montenegro [24,25], Tina U. Nahberger[21], Gabriel F. Peñaloza-Bojacá [26], César Plaza [27], Ana Rey [18,19], Alexandra Rodríguez[19,20], Christina Siebe [28], Alberto L. Teixido[29], Nuria Casado-Coy [5], Pankaj Trivedi [30], Cristian Torres-Díaz[31], Jay Prakash Verma [32], Arpan Mukherjee[32], Xiao-Min Zeng[2], Ling Wang[33], Jianyong Wang [33], Eli Zaady [34], Xiaobing Zhou[35], Qiaoyun Huang [1,2], Wenfeng Tan[2,36], Yong-Guan Zhu[37], Matthias C. Rillig [10,11] & Manuel Delgado-Baquerizo [38,39] ✉

Soil contamination is one of the main threats to ecosystem health and sustainability. Yet little is known about the extent to which soil contaminants differ between urban greenspaces and natural ecosystems. Here we show that urban greenspaces and adjacent natural areas (i.e., natural/semi-natural ecosystems) shared similar levels of multiple soil contaminants (metal(loid)s, pesticides, microplastics, and antibiotic resistance genes) across the globe. We reveal that human influence explained many forms of soil contamination worldwide. Socio-economic factors were integral to explaining the occurrence of soil contaminants worldwide. We further show that increased levels of multiple soil contaminants were linked with changes in microbial traits including genes associated with environmental stress resistance, nutrient cycling, and pathogenesis. Taken together, our work demonstrates that human-driven soil contamination in nearby natural areas mirrors that in urban greenspaces globally, and highlights that soil contaminants have the potential to cause dire consequences for ecosystem sustainability and human wellbeing.

Soil contamination challenges many of the United Nations Sustainable Development Goals such as good health and wellbeing, sustainable ecosystems and cities, and climate change regulation[1,2]. Environmental stress associated with soil contamination, either from natural or anthropogenic origins, can directly affect biodiversity and ecosystem functions[3], and further compromise the resistance and resilience of ecosystems to climate change and natural disasters[1]. Moreover, soil contamination in urban areas can negatively influence the health of city residents through cross-media migration-induced risk (e.g., toxic metal(loid)s affecting drinking water quality and vapor intrusion of

organic contaminants)[4,5]. In addition, urban greenspaces are important recreational places where people have direct contact with soil. Currently, soil contamination is associated with vehicle emissions, industrial processes, weed, and plant disease treatment, as well as poor waste management[1,5]. Thus, urban greenspaces are expected to be more influenced by contaminants than natural ecosystems, which are geographically removed from anthropogenic activities. However, studies have shown that contaminants of concern such as metal(loid)s, pesticides, microplastics, and antibiotic resistance genes (ARGs) can be dispersed through aerial transport, uncontrolled waste disposal/littering, and runoff[6–10], and thus may have impacts on adjacent natural ecosystems. Moreover, some potential soil contaminants have natural origins (e.g., high levels of soil metal(loid)s and ARGs) and can also affect surrounding managed ecosystems. Previous studies have demonstrated significant dispersion of certain contaminants at local and regional scales;[1,7–10] however, these efforts mainly focused on single contaminants across a relatively narrow range of climate and environmental gradients, and many times lack a direct comparison of the most widespread contaminants simultaneously between urban and natural ecosystems. Thus, a global and multidimensional assessment of soil contamination in urban greenspaces is urgently needed to understand the distribution and range of human-driven soil contamination. Furthermore, the importance of human and natural factors in explaining soil contamination across contrasting ecosystems remains virtually unknown.

A global assessment of soil contamination is a huge challenge, as different soil types and climates may have contributed considerable uncertainty to contaminant effects. Yet, any attempt to estimate patterns and consequences of soil contamination requires multiecosystem and multidimensional approaches simultaneously across a broad range of environmental and socio-economic conditions[11]. Moreover, it is imperative to identify how soil functional activities respond to environmental stress typically associated with natural or anthropogenic contaminants. Soil microbes perform critical functions and ecosystem services[12], and are known to respond rapidly to contamination, with changes in the proportion of fundamental microbial traits[3,13]. Because of their environmental sensitivity and widespread distribution, we posit that soil microbes can be used as global indicators of multiple dimensions of natural and anthropogenic soil contaminants (i.e., from microplastics to metal(loid)s). This is vital to better understand the impacts of soil contaminants and will help design strategies to improve ecosystem conservation and human health.

We conducted a global standardized field survey including surface soils collected from 56 paired urban greenspaces and adjacent natural areas (i.e., unmanaged natural/seminatural ecosystems) across six continents (Supplementary. Fig. 1, Table 1). Our paired design allows the direct comparison of urban and natural areas while allowing for biogeographic and macroclimatic patterns, which is a major novelty compared with previous work. We aimed to (i) compare the levels of multiple soil contaminants of concern including metal(loid)s, pesticides, microplastics, and ARGs in urban greenspaces and adjacent natural areas (Supplementary Table 2); (ii) and explore environmental factors associated with soil contaminants in paired areas; and (iii) examine the potential influence of soil contaminants on functional microbial traits associated with soil health such as stress resistance, nutrient cycling, and pathogenesis. Finally, to better understand the importance and implications of soil contamination for the conservation of natural ecosystems, we compared the level of soil contaminants in urban greenspaces with those found in three remote ecosystems from maritime Antarctica (Supplementary. Fig. 2). Our global survey focused on four main groups of soil contaminants of wide concern including eight heavy metals and metalloids (metal(loid)s, hereafter): arsenic (As), cadmium (Cd), chromium (Cr), copper (Cu), lead (Pb), mercury (Hg), nickel (Ni) and zinc (Zn), 46 pesticide residues, the

shape and polymer type of microplastics, and 285 ARGs (Supplementary Table 3). In the case of metal(loid)s and ARGs, we acknowledge that they are also naturally enriched in certain soils, irrespective of contamination. Therefore, to interpret the degree to which values of these contaminants can be attributed to anthropogenic activities, we identified the associations between natural (climate, plants and soil) and human factors (population size and density, human development index (HDI), gross domestic product (GDP), and management practices of urban greenspaces) and soil contaminants. Further, we explored the links between the co-occurrence of multiple soil contaminants and microbial functional attributes (e.g., genes associated with biological stress), which remain unexplored and are critically important to predict potential effects of different widespread contaminants on soil functions.

## Results and Discussion
Our survey demonstrates that, irrespective of their origins (geochemical background vs. anthropogenic), soil contaminants (i.e., metal(loid)s, pesticides, microplastics, and ARGs) are widely distributed in terrestrial ecosystems. In particular, we found similar levels of multiple dimensions of soil contamination in urban greenspaces and adjacent natural areas across continents (Supplementary Fig. 3 and Supplementary Table 4). The same results were obtained when using a response ratio of soil contaminants (urban vs. natural; Fig. 1). Although the level of individual contaminants varied greatly across locations, we detected significant correlations among each type of soil contaminants studied (Supplementary Fig. 4). Our data further provide a snapshot of multiple soil contaminants in natural areas, comparing with the adjacent urban greenspaces worldwide (Supplementary Table 5 and Supplementary Fig. 5). We also found significant levels of these contaminants in the soil of remote Antarctica, confirming earlier local reports of atmospheric transport of specific contaminants including pesticides and microplastics[14,15]. Moreover, we provide evidence that human-associated factors are essential to explain the distribution of multiple soil contaminants across contrasting ecosystems (Fig. 2), supporting an important link between anthropogenic activity and soil contamination that goes beyond the natural origin of particular contaminants, such as the case of metal(loid)s. Finally, we found that levels of soil contaminants were significantly correlated with the relative abundance of important functional genes associated with stress resistance (i.e., the resistance to stresses of metal(loid), drugs, and pathogens), nutrient cycling, pathogenesis, and microbial metabolism (Fig. 3). Importantly, we found that an increasing number of soil contaminants can have a greater contribution in explaining the proportion of microbial functional traits (see Supplementary Fig. 6). Our work is relevant because it provides evidence of a quantitative comparison of soil contaminants in urban greenspaces and adjacent natural areas across six continents. The results revealed that adequate conservation and functioning of terrestrial ecosystems requires an assessment of risk associated with multiple dimensions of soil contamination.

### Multiple dimensions of soil contaminants in paired urban and natural areas
We found similar levels of metal(loid)s such as Pb, Ni, Cd, and As in soils of urban greenspaces and adjacent natural areas, but urban greenspaces had higher contents of soil Hg, Zn, Cu, and Cr than those in natural ecosystems (Supplementary Fig. 3A). Differences in the accumulation of metal(loid)s in urban greenspaces compared with natural areas could be related to geographic variations in the extent of metal(loid) emissions from human activities, though the exact mechanisms remain unknown[16]. The broad spectrum of anthropogenic impacts considered in this study (e.g., Pb and Hg) captures, at least partially, the potential anthropogenic sources of these elements. However, the natural sources and geochemical background of these

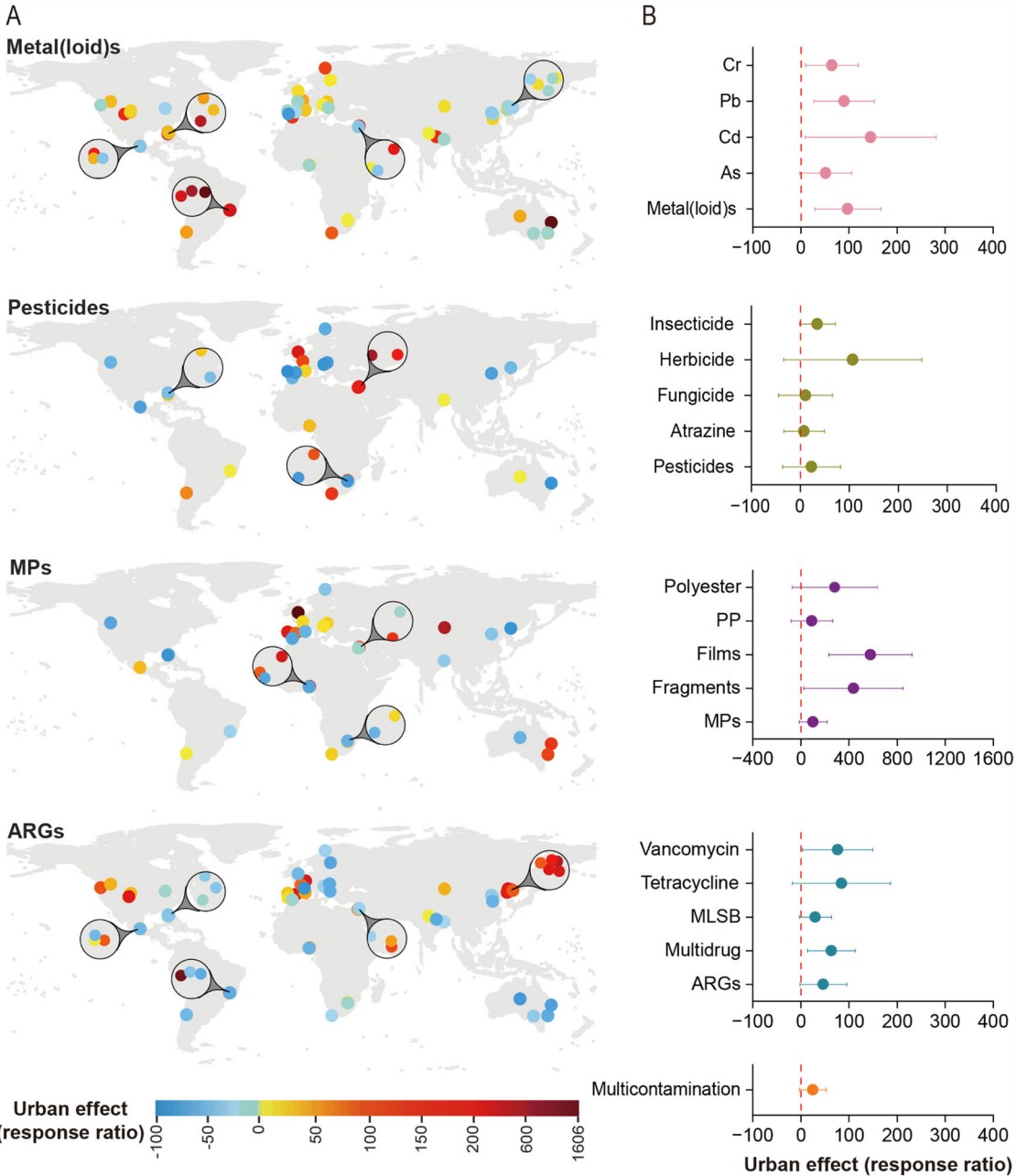

**Fig. 1 | Changes (%) in soil contaminants in urban greenspaces compared with adjacent natural areas worldwide. A** response ratios of (site-level described in Methods) of standardized contaminant indices of metal(loid)s, pesticides, microplastics (MPs), and antibiotic resistance genes (ARGs). **B** the changes (site-level response ratios, mean ± 95% confidence interval) in the selected soil contaminants within the four categories (i.e., metal(loid)s, pesticides, MPs, and ARGs) and multi-contamination (represented by different colors), comparing urban greenspaces to adjacent natural areas (see Supplementary Fig. 3 for the concentrations of soil contaminants). Multi-contamination represents an average contamination index of standardized four categories of soil contaminants. Complementary figures showing results using an independent approach can be seen in Supplementary Fig. 3, 7 and 10, 11. The maps in the figure were generated using ArcGIS 10.2 software.

trace metals are also critical in explaining levels of these metal(loid)s[17]. Metal(loid)s in the topsoils of urban greenspaces may affect soil organisms and ecosystem processes when they exceed a specific threshold concentration. For instance, according to, the guidelines of the Finnish legislation for soil contamination[18], standards that are widely used to assess soil contamination, 42% of urban greenspace sites and 36% of sites in natural areas exceeded the lower limit for As contamination in soil (Supplementary Table 6). As expected, however, average levels of these metal(loid)s in the remote Antarctica ecosystem were below the legislation limit (Supplementary Figs. 7 and 8). These findings advance our knowledge of the typical levels of toxic

metal(loid)s found in urban greenspaces and adjacent natural areas worldwide. Accumulated contaminants in surface soils of greenspaces do not necessarily increase environmental risk because the bioavailability of soil contaminants depends on soil properties such as pH, texture, and organic carbon content[5].

Similarly, pesticide residues were widely detected in many urban greenspaces and natural ecosystems at a global scale. We detected the presence of pesticide residues, including fungicides, herbicides, and insecticides, in 63% of the surveyed natural ecosystems (Supplementary Fig. 3B; Supplementary Table 6). Our analyses likely underestimate these levels as we did not measure several widely used pesticides, such

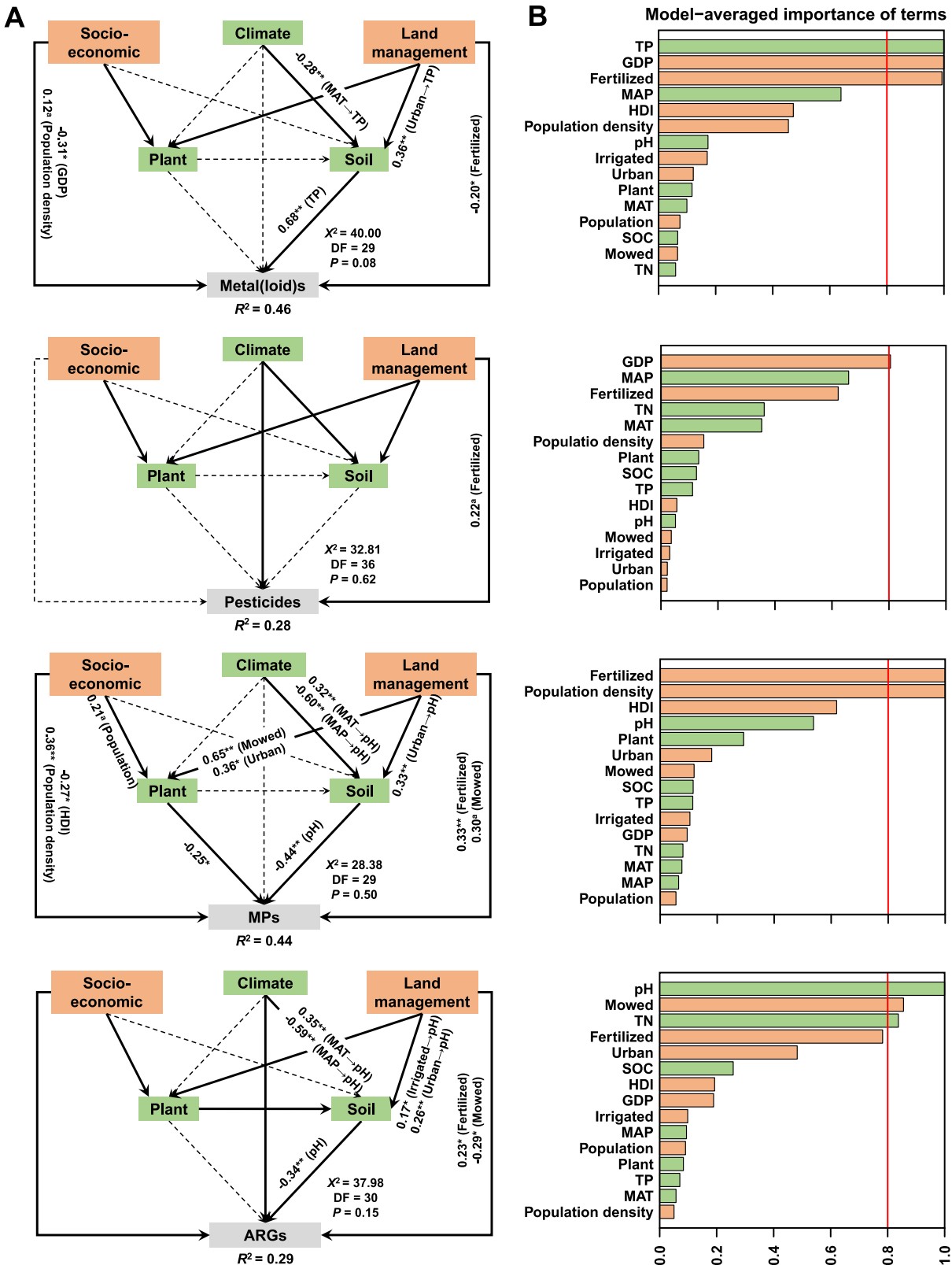

**Fig. 2 | The relative importance of socio-economic and environmental factors on the accumulation of soil contaminants. A** structural equation models reveal direct and indirect associations of examined socio-economic (population size and density, human development index (HDI), gross domestic product (GDP)) and environmental factors (climate, plant cover, and soil) with indices of metal(loid)s (*n* = 112 plots), pesticides (*n* = 54 plots), microplastics (MPs) (*n* = 64 plots) and antibiotic resistance genes (ARGs) (*n* = 112 plots) in the soil accounting for different management practices (irrigated, mowed and fertilized). Numbers adjacent to arrows are indicative of the effect size of the relationship. **B** the relative importance of examined human and natural factors on the four groups of soil contaminants, based on linear mixed effects model. Orange and green boxes indicate human and natural factors, respectively. MAT mean annual temperature, MAP mean annual precipitation, SOC soil organic carbon, TP total phosphorus, TN total nitrogen. Urban urban greenspaces, Plant plant cover.

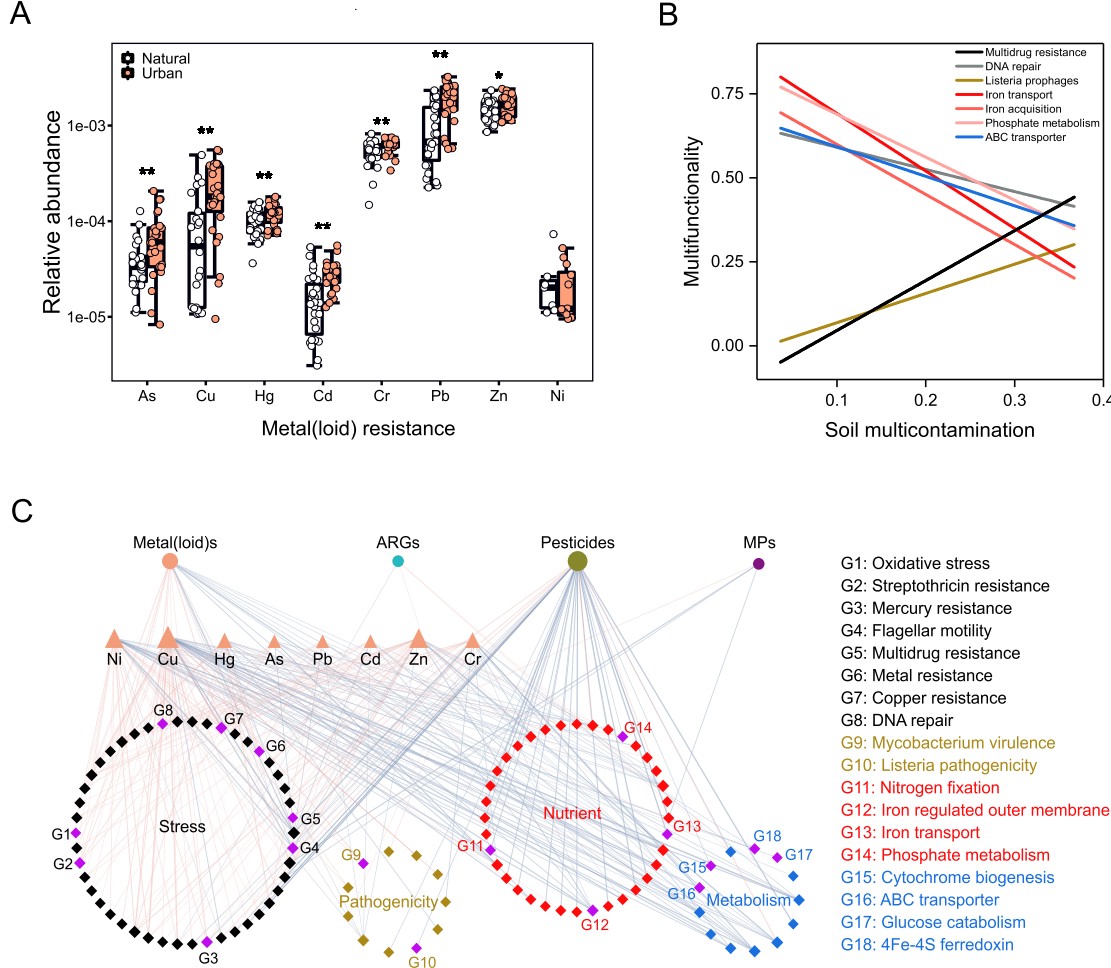

**Fig. 3 | Associations between soil contaminants and functional soil microbiome. A** proportions of the genes associated with metal(loid) resistance. Asterisks indicate significant differences between urban and natural ecosystems according to nested PERMANOVA using a block design as described in the Method section. *$P < 0.05$; **$P < 0.01$. Detailed statistical results for the PERMANOVA are shown in Supplementary Table 8. **B** the fitted linear relationships between multi-contamination (standardized between 0 and 1, see Methods) and proportions of the selected functional genes, which are related to stress resistance (black), nutrient cycling (red), pathogenesis (green), and microbial metabolism (blue). Statistical analysis was performed using ordinary least squares linear regressions; $P < 0.05$; $n = 32$. **C** a correlation network displaying relationships between soil contaminants and selected functional genes associated with well-known stress resistance, pathogenicity, nutrient cycling and microbial metabolisms. These functional genes were annotated according to BacMet and MG-RAST using meta-genomic data (see Methods and Supplementary Table 9). MPs microplastics, ARGs antibiotic resistance genes.

as glyphosate, glufosinate, paraquat, and 2,4-Dichlorophenoxyacetic acid. We did not detect a difference in the occurrence of pesticides between urban and natural greenspaces (Supplementary Fig. 3). This is also evident when analyzing response ratios (urban vs. natural; Fig. 1B) or accounting for our paired design (Supplementary Fig. 3). The application of persistent pesticides in urban areas may influence adjacent natural areas, for example through surface runoff and atmospheric transport and deposition[14]. In addition, agricultural pesticide application may contribute to the reported levels of pesticide residues in adjacent urban greenspaces and natural areas by deposition.

Microplastics, typical anthropogenic-driven contaminants, were also ubiquitous in soils of both urban greenspaces and natural ecosystems worldwide. Strikingly, we found similar contents of soil microplastics in the paired ecosystems worldwide (Fig. 1; Supplementary Fig. 3C). The average level of microplastics was 916.2 items kg⁻¹ soil (range from 166.7 to 3482.3 items kg⁻¹), comparable to previously reported values[19,20]. The detected microplastics consisted predominately of polypropylene and polyester, with fibers as the dominant shape (Supplementary Fig. 9), consistent with reports of microplastic levels in the ocean and terrestrial ecosystems[21]. We also found similar

proportions of the shape and polymer type of microplastics in natural areas and urban greenspaces (Fig. 1B; Supplementary Fig. 10), which further supports the idea of a spillover of anthropogenic contaminants across ecosystems. These microplastics, which typically originate from cities, can also reach distant areas by atmospheric transport, with fibers being the main shape of airborne plastic particles in the atmosphere of cities such as Paris[22], London[23] and Dongguan (China)[24]. These fibers generally consist of polyester and polypropylene, which can originate from synthetic fabric, rope, and nets. In addition, semi-natural ecosystems can also accumulate microplastics due to human activities, leading to waste disposal /littering, runoff, and transport through the atmosphere[25]. Unexpectedly, our results showed that soils in remote Antarctica had similar signatures of microplastics (in both concentration and type) in surface soils to our surveyed urban greenspaces worldwide (Supplementary Fig. 7). This could be related to microplastic dispersion from Antarctic research stations and other continents by sea and air, though this would depend on wind strength and direction[26]. For example, recent studies showed that more than 60% of microplastics detected in Antarctic samples were <50 μm, and these tiny-sized plastics can be transported over long distances before deposition[27,28]. Other activities, such as, tourism may also contribute to the accumulation of

microplastics in the soils of Antarctica sites[29]. Future studies should provide more detailed measurements of the characteristics of microplastics in order to track their distribution and fate in the environment. On the whole, our findings provide evidence that the level and characteristics of microplastics in natural areas match those in urban city parks and gardens, with multiple conservation and ethical implications for microplastic contamination of terrestrial ecosystems worldwide.

We then characterized the richness and abundance of 285 ARGs in the soils from the 112 ecosystems studied (Fig. 1; Supplementary Fig. 11). Despite their natural origin (soil-borne bacteria), soil ARGs are emerging as potential biological contaminants threatening human health[6,30], and the presence of some ARGs in soil has been linked to the widespread use of antibiotics in medical and agricultural industries (e.g., veterinary medicine)[10]. The intense presence of pet excreta, atmospheric deposition, and population movement may be the main transfer routes of ARGs to the soil of urban greenspaces[10,31]. Thus we consider them an indirect measure of antibiotic influences in highly disturbed environments such as urban greenspaces. Our results show the ubiquity of ARGs in both urban greenspaces and natural areas, though the proportion of specific ARG types varied across locations (Supplementary Figs. 5 and 11). The similarity in ARGs between natural and urban soils might be related to the physical mobility of bacterial cells and plasmids carrying ARGs from urban to natural environments[32]. We show that soils from urban greenspaces supported a greater diversity (i.e., richness) of ARGs than natural ecosystems, suggesting that urban areas may receive more diverse sources of ARGs from anthropogenic sources. These analyses advance our current understanding of ARGs by providing quantitative PCR data to investigate the diversity of ARGs in urban environments[33], though our analyses may not include those ARGs that are below detection limits. In addition, the significantly lower richness of ARGs in remote Antarctica (Supplementary Fig. 7) further suggests that the presence of ARGs could be linked to human antibiotic use.

## Human and natural factors associated with soil contamination across the globe

To provide further insights into their origin and anthropogenic contribution to soil contamination in urban and natural ecosystems, we investigated the association among socio-economic and environmental factors and soil contaminants. Structural equation modelling (SEM) and linear mixed-effects models consistently revealed that regional population density was the most important socio-economic factor associated with microplastics (Fig. 2A, B; Supplementary Figs. 12 and 13). This finding accords with local studies showing significant links between population density and environmental contamination[34,35]. Densely populated areas are typically associated with greater microplastic inputs from traffic, industrial emissions, domestic waste and the release of synthetic fibers[5,36], or sewage sludge and compost used as soil fertilizers[37]. We also found negative associations between city wealth (reflected in GDP and HDI, Supplementary Fig. 14) and soil microplastics and metal(loid)s (Fig. 2; Supplementary Fig. 12). Significant associations between response ratios of soil contaminants and population density further underline the importance of human factors in the accumulation of soil contaminants, particularly microplastics in urban ecosystems (Supplementary Table 7). According to our modelling analyses, human factors such as soil fertilization played more important roles than natural factors in influencing the abundance of pesticides and ARGs worldwide (Fig. 2B). For instance, despite the natural origin of ARGs, fertilization was positively associated with the abundance of ARGs (Fig. 2). The widespread use of sewage sludge and animal manure containing antibiotic residues may have increased the presence of ARGs in fertilized urban sites[31]. Taken together, we posit that human activities are causing a wide spillover of multiple soil contaminants across ecosystems, transporting contaminants between urban greenspaces and natural areas by atmospheric deposition and surface runoff. Soil contamination, however, is a complex process, and may be affected by other undocumented factors, many of which act at a local scale. Our statistical analyses are also limited in that we cannot assume causal relationships among these factors. Long-term field monitoring and controlled experiments are needed to strengthen our understanding of the diffusion and accumulation of soil contaminants across contrasting ecosystems. Our work points to the global magnitude of these processes and their links to human activities.

## Associations among soil contaminants and microbial functional traits

Finally, we used metagenomic data to establish potential associations among multiple soil contaminants and biological responses associated with key functional aspects maintaining ecosystem sustainability[33]. We found, for example, that greater contents of soil Cr, Cu, Hg and Zn in urban greenspaces than in natural areas matched the larger proportion of genes associated with biological resistance (e.g., multidrug resistance) in urban environments (Fig. 3A; Supplementary Fig. 15, Table 9). Moreover, the proportion of genes associated with resistance to streptothricin (an antibiotic) was positively correlated with elevated concentrations of Hg across soils (Supplementary Fig. 16). These findings suggest that the soil functional microbiome is linked to metal(loid) concentrations. In addition, other well-known genes associated with stress resistance, nutrient cycling, pathogenesis and microbial metabolism were associated with a greater level of multiple soil contaminants when considered simultaneously (multi-contamination calculated as the averaged of standardized concentration of contaminants, see Methods) (Fig. 3B, C). For example, the soil multi-contamination index was positively correlated with the proportion of the genes associated with pathogenic *Listeria prophages* and multidrug resistance, but negatively correlated with the genes involved in P metabolism, Fe acquisition and transport, and DNA repair (Fig. 3B). These results suggest that the accumulation of contaminants potentially increased the pathogenicity of soil microbes and affected their functions. We also found negative associations among pesticides and genes associated with nutrient (e.g., N, P, and Fe) cycling and basic metabolisms (e.g., ABC transporter and 4Fe-4S ferredoxin) (Fig. 3C; Supplementary Figs. 16 and 17). Importantly, we found that the number of soil contaminants over certain thresholds was significantly correlated with the proportions of the selected functional genes (Supplementary Fig. 6). These results indicate significant impacts of multiple contaminants on the soil functional microbiome, which is critical to supporting ecosystem services[12]. However, gene abundance does not necessarily reflect functional activity[38], and further experiments are needed to improve our understanding of soil contamination impacts on functional attributes supported by the soil microbiome. Experimental studies manipulating soil contaminants and testing how they influence soil health are important to test further links between soil contaminations and functional gene activities observed here. Such knowledge will help to protect ecosystem functions from human-driven contamination across the globe efficiently.

## Implications

Our study provides the cross-continental-scale assessment of the patterns and biological responses associated with soil contaminants in natural and urban areas. We provide evidence that soil contamination in natural areas surrounding cities mirrors that in urban greenspaces worldwide. This highlights the growing global contaminant footprint, which has reached even remote locations such as Antarctica. We also found that socio-economic factors are significantly associated with the occurrence of soil contaminants, with positive associations between population density and the levels of metal(loid)s and microplastics, highlighting the contribution of humans to soil contamination regardless of their origins. We further reveal the potential influence of

soil contamination on key microbial traits such as stress resistance, nutrient cycling, and pathogenesis. These findings pave the way to the formulation of hypotheses that can be tested using controlled laboratory- and field-based studies. Together, our work demonstrates that soils in nearby natural areas are as contaminated as our urban greenspaces at a large-spatial scale. Future studies on a wider spectrum of contaminants (e.g., pharmaceuticals) in paired ecosystems across more specific regions should consolidate our understanding of soil contamination and its environmental risks.

## Methods

### Field survey and soil sampling
Soil samples were collected from paired urban greenspaces and adjacent natural areas (i.e., natural and semi-natural ecosystems near cities that are not subjected to management) from 56 municipalities of 17 countries and six continents (Supplementary Fig. 1A, B and Supplementary Table 1). Natural areas were about 20 km from the urban greenspaces, which include forests, grasslands, shrublands, or relict forests with their original vegetation. Urban greenspaces were mainly public parks and large residential gardens, and comprised a mixture of open areas with lawns, scattered trees, patches of shrubs, and associated flowerbeds. All the selected urban greenspaces were well-established and many decades old. Our findings indicate that soils in urban and natural ecosystems were similar in their levels of contamination. A total of 112 ecosystems (56 paired urban vs natural areas) were investigated in this study. The mean annual precipitation and temperature ranged from 210 to 1577 mm and 3.1 °C to 26.4 °C, respectively. These natural areas are locations with no historically documented or evident human impact. The selected cities have a wide range of socio-economic development indices, representing different development levels of cities worldwide. For example, region population density and human development index (HDI, a summary measure of average achievement in key dimensions of human development)[39] ranged from 7.7 to 16 people/km$^2$ and 0 to 0.89, respectively. We did not sample agricultural soils, and the selected sites were remote from any mining areas. We also considered soil contamination attributes and sampling feasibility when choosing sampling sites across such a broad range of environmental gradients worldwide.

A 30 m × 30 m plot (900 m$^2$) consisting of three parallel transects of equal length was surveyed at each location[33]. These plots were selected to represent the most common environments within ecosystems (e.g., a grass lawn or an urban forest in a city park). We then collected surface soils (top 5 cm) from the 112 ecosystems. To account for spatial heterogeneity, we collected three sampling points for a composite sample under the most common environments (vegetation and open areas between plant canopies covered by bare soils and nonvascular plants) found at each site (Supplementary Fig. 1C). Thus we obtained 336 composite soil samples in total. For comparison, we collected three additional composite soil samples (from three sampling points) from Antarctica, which is far from human activities. After field collection, soils were sieved (<2 mm) for downstream analyses. The sieves were carefully washed, and cleaned with 90% ethanol between samples to avoid cross- contamination. One sub-sample was air-dried for chemical (i.e., pH, organic carbon, total phosphorus, and nitrogen, metal(loid)s, pesticide residues, and microplastics) analyses, and the other soil fraction was frozen at −20 °C for the analyses of microbiome including ARGs.

Due to sample availability and resource limitation, we did not analyze all soils for each category of contaminant. Many of the analyses included in this study (e.g., polymer identification of microplastics and 46 pesticide residues) are highly costly and time-consuming, which require a relatively large amount of sample for contaminant extraction (see Methods below). We thus focused on the subset of sites potentially impacted by specific contaminants according to advice from local greenspace managers and environmental scientists. Therefore,

we attempted to cover the entire gradient of conditions by focusing on a subset of the samples. Thus, we measured metal(loid)s and ARGs in all the 336 composite samples from 112 plots, microplastics in 64 composite samples from the selected 64 plots and pesticide residues in 54 composite samples from the selected 54 plots (See Supplementary Fig. 1, Table 3). For comparability, we standardized within-site replicates for metal(loid)s and ARGs (e.g., 112 plots × 3 composite samples = 336 samples). In all cases, the subsets of samples were selected to cover the entire biogeographic range along with a broad range of environmental gradients (Supplementary Fig. 1 and Table 2). To the best of our knowledge, this is the largest paired sampling dataset of soil contamination worldwide. We acknowledge, however, the potential limitations of an unbalanced number of samples for the analyses of the four categories of soil contaminants.

### Environmental factors
Climatic information (i.e., mean annual precipitation and temperature) were extracted from the WorldClim database (https://www.worldclim.org/data/index.html), and averaged values of climatic parameters from recent years (2013–2021) were used in this study. Plant cover was determined at each site based on three transects across 30 m × 30 m plots. Information on the population was obtained from the latest available city censuses using official national statistical sources[33]. GDP and HDI per capita (over 1990–2015) in the regions for the cities surveyed were extracted from the reported dataset[40], providing information on the economic activity and key dimensions of human development for each location. Moreover, we collected information about mowing, irrigation, and fertilization treatments by surveying the managers of urban greenspaces[33].

### Analyses of soil chemical properties
Soil pH was determined on a 1: 5 soil/water extract using a pH electrode. Total carbon and nitrogen in the soil were analyzed using an elemental analyzer (C/N Flash EA 112 Series-Leco Truspec). Soil organic carbon was measured using the same elemental analyzer after fumigation with HCl[41]. Total phosphorus was determined using an inductively coupled plasma optical emission spectrometer (ICAP6500 DUO; Thermo-Scientific) after digestion using nitric-perchloric acid.

### Analyses of soil contaminants
For the analyses of Cu, Pb, Cd, Zn, Ni, Cr, and As, 0.25 g of each finely ground (<0.149 mm) soil was digested by a MARS microwave digestion system using mixed acids (9.0 mL of HNO3 and 3.0 mL of HF)[16]. The concentrations of metal(loid)s in the digestion were subsequently determined by Inductively Coupled Plasma-Optical Emission Spectrometry (ICP-OES) with Thermo ICP 6500 Duo equipment (Thermo Fisher Scientific, Waltham, MA, USA). In parallel, 0.20 g of each finely ground soil was digested with 3:1 *aqua regia* for total Hg analysis, and the amount of Hg in the digestion was measured using cold vapor atomic fluorescence spectrometry (CVAFS)[38]. Analytical blanks were included in all analyses for quality control. The limit of detection in for each element was (in µg g$^{-1}$): 0.0008 for As, 0.0006 for Cd, 0.0011 for Cr, 0.0012 for Cu, 0.0011 for Ni, 0.0008 for Pb, 0.0014 for Zn and 0.0001 for Hg.

The pesticide residues in soils were extracted and measured as a previous method[42]. Briefly, Accelerated Solvent Extraction (Dionex ASE 350, Thermo Scientific) was used to extract pesticide residues from 6 g of soil. The extraction contained two steps. In the first step, an organic mixture of acetone, methanol, and acetonitrile at a ratio of 65:10:25 (% v/v) was used. In the second, pesticide residues in soil were extracted with a mixture of acetone and 1% phosphoric acid in Millipore water at a ratio of 70:30 (% v/v). The extracts were processed, and the pesticide residues were analyzed by high-performance liquid chromatography coupled to a triple quadrupole tandem mass spectrometer (HPLC-MS/MS). Reversed-phase

HPLC using water and methanol as mobile phase was used to perform the chromatographic separation. Detection was performed with a triple quadrupole mass spectrometer (QTrap 5500, Sciex) and all detected concentrations ($\mu g\,L^{-1}$) were converted into µg per kg of dry soil. A total of 46 different pesticide residues were assessed. The limit of quantification ranged from $0.064\,\mu g\,kg^{-1}$ to $36\,\mu g\,kg^{-1}$ depending on the substance. We want to acknowledge that the analytical method was not designed to cover all pesticide residues potentially present in soil, but still provide a good overall view of the soil contamination associated with pesticides. Specifically, out of the 20 worldwide most applied active pesticides ingredients, our method can detect up to 30% of these substances, including the quantification of the frequently used active pesticide ingredients such as acetochlor, alachlor, atrazine, azoxystrobin, chlorpyrifos, clothianidin, diuron, imidacloprid propiconazole, S-metolachlor tebuconazole, and trifloxystrobin.[43]

Microplastics (i.e., plastics <5 mm in diameter) in soil were pretreated using Fenton's oxidization method, while maintaining the integrity of the plastic particles.[44,45] The Fenton's reagent was added slowly in small quantities to minimize the potential effect on the recovery of plastics. Ice bath was used to maintain temperatures below 40 °C. Samples were pretreated with Fenton's reagent to oxidize organic matter for 5 hours. After that, the wall of the beakers was rinsed with ultrapure water to detach any microplastics and left to dry (at 45 °C) for 1 day. Microplastics were then extracted through density separation with a saturated NaCl solution, and the soil was stirred with a glass rod for 3 min and allowed to settle overnight. To improve the recovery rate of microplastics, we repeated the extraction of each soil sample four times and used calcium bicarbonate to promote flotation via the release of $CO_2$ bubbles[36]. The supernatant of samples was subsequently decanted, filtered and stored for the next analysis[46]. Filters were analyzed and counted visually using a binocular microscope glass (MOTIC SMZ 143 N2GG, Wetzlar, Germany) to find all the tentative microplastic particles, ranging from 5 mm to 20 µm. The shape (i.e., fiber, fragment, and film) of the particles was assessed according to a previous method[47]. Noted that this method may have underestimated actual microplastic levels due to limitations in the extraction method. A representative number of particles were subsequently analyzed for the identification of polymer type by RAMAN spectroscopy (NRS-5100, Jasco, Madrid, Spain) equipped with LMU-20X-UVB lens. The laser excitation frequency and intensity used was 784.79 nm and 11.8 mW, respectively. Raman spectra were obtained between 162 and 1886 $cm^{-1}$ with a spectral resolution of 2.47 $cm^{-1}$, and were recorded with a charge-coupled device camera (UV-NIR range, 1024 × 255 pixels) electrically cooled to −70 °C. The RAMAN spectra of the analyzed particles were finally compared to reference polymers from the spectral library Open Specy[48].

Microplastics extracted from the soils were checked to ensure they were not contaminated by plastic bags used in the collection and storage processes. We determined the spectroscopic signature of all the bags used for the sampling, which did not match any of the signature of microplastics found in the samples. To minimize microplastic contamination of samples throughout the laboratory procedures, the plastic material was avoided as much as possible, using glass material that had been cleaned with ultrapure water, cotton laboratory coats and, whenever possible, clothes that were made of natural fibers. The NaCl solution was always previously filtered to remove any possible microplastics in the salt[49]. Additionally, containers holding the samples were covered to prevent airborne contamination of microplastics. To control for possible cross-contamination, procedural blanks were used during all the procedures in the laboratory and no microplastics were detected in them[50].

Genomic DNA from soils of urban greenspaces and natural areas was extracted using the PowerSoil DNA Isolation Kit (QIAGEN Inc., Germany) following the manufacturer's instruction. A high-throughput quantitative PCR (HT-qPCR) based chip was used to quantify ARGs on the Wafergen SmartChip Real-Time PCR System (Fremont, CA, USA). The ARG chip consisted of 285 primer sets targeting ARGs conferring resistance to all major classes of antibiotics, including aminoglycoside, beta-lactams, FCA (fluoroquinolone, quinolone, florfenicol, chloramphenicol and amphenicol), macrolide-lincosamide-streptogramin B (MLSB), multidrug, sulfonamide, tetracycline and vancomycin (Supplementary Table 3). Meanwhile, the 16S ribosomal RNA gene was included as a reference gene. All HT-qPCR reactions were performed in technical triplicates with a negative control.[51]

### Analyses of microbial functional traits

In total, 54 composite topsoil samples (one composite sample per plot) from 27 typical paired urban and natural areas were selected for metagenomic analyses. Sufficient amounts (roughly 500 ng) of microbial DNA were extracted using the DNeasy PowerSoil DNA Isolation Kit (QIAGEN Inc., Germany) according to the manufacturer's protocol. Shotgun sequencing was performed using an Illumina HiSeq (Illumina Inc., USA) at Majorbio in Shanghai, China. Raw reads (PE150, 150 bp pairedend reads) were trimmed to remove low quality reads. The SeqPrep software (https://github.com/jstjohn/SeqPrep) was used to remove adapter sequences. Then, the library sickle (https://github.com/najoshi/sickle) was used to trim the reads from the 5′ end to 3′end using a sliding window (size 50 bp, 1 bp step). All reads below the quality threshold of 20 were trimmed. The resultant reads below 50 bp or containing N (ambiguous bases) were discarded. The trimmed reads were then mapped against the protein sequence of the BacMet (http://bacmet.biomedicine.gu.se/) and Subsystem Technology (MG-RAST; https://www.mg-rast.org)[52], respectively. MG-RAST generates taxonomic assignments based on the SEED subsystem database by DIAMOND software (version 0.9.32) by best-hit classification with a maximum E-value of 1e$^{-5}$, a minimum identity of 60%, and a minimum alignment length of 25 amino acids for proteins and functional categories. The resulting table was parsed at SEED Subsystem Level 3 by the software SUPER-FOCUS. The relative abundances of annotated genes were calculated using the number of reads mapping to genes for each sample. We focused on the genes associated with ecosystem functions such as the resistance to environmental stress, pathogenesis, nutrient cycling and microbial metabolism. The resistance traits included the resistant genes to stress of metal(loid)s, drugs, and pathogens. Detailed information on microbial traits was assessed by the gene abundances in the soil microbiome is shown in Supplementary Table 9.

### Statistical analyses

We first calculated a soil contaminant index for each category by averaging standardized values of the individual soil contaminants. To further provide an overall assessment of soil contaminants, we calculated the multi-contamination index by averaging standardized values of four categories of examined soil contaminants including metal(loid)s, ARGs, pesticide residues and microplastics. We used the following standardization approach so that all variables contribute similarly to the multi-contamination indices (1):

$$standardization = \frac{[variable\ content - \min(variable\ content)]}{[\max(variable\ content) - \min(variable\ content)]} \quad (1)$$

variable is the level of the examined soil contaminant from a specific site, and the min (variable content) and max (variable content) represent the minimum and maximum level of each soil contaminant among all the examined sites. Thus, each transformed variable had a minimum value of zero and a maximum value of one. This approach is similar to the methods used for calculating ecosystem multifunctionality[3,53]. It is noted that only the sites simultaneously having data for four categories of soil contaminants were selected to

calculate the multi-contamination index ($n = 48$ sites). Levels of metal(loid)s and ARGs derived from three composite soil samples per plot were averaged to obtain plot-level estimations prior to statistical analyses[54,55]. This allowed us to relate the spatial heterogeneity within plots to the metrics of soil contaminants.

We calculated response ratios[33] of contaminants in the urban greenspaces vs. adjacent natural ecosystems. Response ratios were used to assess the enrichment of examined soil contaminants in urban greenspaces by comparing them with the adjacent natural ecosystems as follows (2):

$$\text{Response ratio} = \frac{\text{Value variable A urban} - \text{Value variable A natural}}{\text{Value variable A natural}} \times 100 \quad (2)$$

We also used a blocked design to test differences in contents of contaminants (i.e., metal(loid)s pesticides, microplastics, and ARGs) between urban greenspaces and the adjacent natural ecosystems accounting for paired locations (i.e., we fixed location in the strata), providing complementary statistical support to the response ratios. These analyses were performed using nested PERMANOVA (permutations = 999) in the R package "Vegan" ("adonis" routine) in R 4.0.3.

We used structural equation modelling (SEM)[56] to identify the relative importance of socio-economic vs. environmental factors in explaining the accumulation of four major types of contaminants (i.e., metal(loid)s, microplastics, pesticide residues and ARGs) from the natural and urban ecosystems. An a priori model was established based on our current knowledge regarding environmental impacts on soil contamination (Supplementary Fig. 18). These factors include socio-economic (i.e., population size and density, HDI and GDP), soil factors (i.e., pH, and organic carbon, total phosphorus and nitrogen), plant cover, climate (mean annual precipitation and temperature) and land management (i.e., fertilization, irrigation and mowing of urban greenspaces). We assume that socio-economic factors, land management and climate generally influence plant and soil variables, and therefore directly and indirectly affect the accumulation of soil contaminants. For example, countries with a high GDP generally invest more to waste management and have more restrictive policies in terms of environment protection, compared with countries with low GDP. The management practices were included in our SEM as categorical variables with two levels: 1 (a given management practice) and 0 (no management practice). Information on variables included in the model is either measured in situ or in the laboratory, or by collecting data from the literature as explained above. The SEMs allowed us to determine the direct and indirect effects of multiple environmental factors on soil contaminants. The probability that a path coefficient differs from zero was tested using bootstrap resampling. Doing this instead of assessing significance with frequentist methods allowed us to overcome limitations related to the data matching a particular theoretical distribution. The data matrix was fitted to the model using the maximum-likelihood estimation method. There is no single universally accepted test of the overall goodness of fit for SEM. Thus, we used the Chi-square test ($\chi^2$; the model has a good fit when $0 \leq \chi^2/\text{d.o.f} \leq 2$ and $0.05 < P \leq 1.00$) and the root mean square error of approximation (the model has a good fit when $0 \leq \text{RMSEA} \leq 0.05$ and $0.10 < P \leq 1.00$). The SEM analyses were performed using AMOS 21.0 (SPSS Inc., Chicago, IL, USA).

In parallel, we evaluated the relative importance of individual natural and human factors on the four types of soil contaminants by a mixed-effects model using the "glmulti" package in R. The importance of each predictor was expressed as the sum of Akaike weights for models. We used a cutoff of 0.8 to differentiate between essential and nonessential predictors as done in a previous study[57]. We also used Spearman's correlation analyses to evaluate associations between selected factors and soil contaminants and the response ratios.

We explored associations between potential soil contaminants and the selected functional genes, and observational correlations are essential to understand ecosystems and develop new hypotheses[3]. For simplicity and research focus, we conducted ordinary least square linear regressions between soil multi-contamination index and the proportions of the genes associated with environmental stress, pathogenesis, nutrient cycling and basic metabolisms. To strengthen our understanding of soil contaminant impacts, we analyzed the relationships between the number of soil contaminants over four thresholds of maximum contaminant levels and the relative abundances of selected functional genes (see Supplementary Method for details). In addition, we used a co-occurrence network approach to visualize the associations between the soil contaminants (i.e., metal(loid)s, pesticides, microplastics and ARGs) and selected functional genes. To achieve this, Spearman correlations between soil contaminants and these functional genes were calculated using R package "psych", and we considered $P < 0.05$ as statistical significance. The network was then visualized using the Cytoscape software. We also evaluated the relationships between specific soil contaminants and the proportions of these functional genes.

### Reporting summary
Further information on research design is available in the Nature Portfolio Reporting Summary linked to this article.

## Data availability
The data that support the findings of this study publicly available at figshare: https://figshare.com/articles/dataset/DATASET_Yu-RongLiu_20230216/22107971; https://doi.org/10.6084/m9.figshare.22107971 (ref[58].).

## Code availability
All code associated with our analyses in this study is available at https://figshare.com/articles/dataset/Rscript_Yu-RongLiu_20230216/22107818; https://doi.org/10.6084/m9.figshare.22107818.

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

## Acknowledgements
We thank the researchers involved in the MUSGONET project for collection of field data. This study was supported by a 2019 Leonardo Grant for Researchers and Cultural Creators, BBVA Foundation (URBANFUN), and by the BES grant agreement No LRB17\1019 (MUSGONET). We are grateful for the assistance of Yunyun Hao and Xuemei Han during soil sampling. We also thank Drs. Shuai Du and Xiuli Hao for their help in data analyses. M. D-B. is supported by the projects from the Spanish Ministry of Science and Innovation (PID2020-115813RA-I00) (SOIL4GROWTH) and TED2021-130908B-C41 (URBANCHANGE) funded by MCIN/AEI/ 10.13039/501100011033, and a project of the Fondo Europeo de Desarrollo Regional (FEDER) and the Consejería de Transformación Económica, Industria, Conocimiento y Universidades of the Junta de Andalucía (FEDER Andalucía 2014-2020 Objetivo temático "01 - Refuerzo de la investigación, el desarrollo tecnológico y la innovación") asociado with the research project P20_00879 (ANDABIOMA). Y-R. L. is supported by the National Natural Science Foundation of China (42177022). M.G.A.H is supported by the Swiss National Science Foundation (310030_188799). D.J.E. is supported by the Hermon Slade Foundation. F.B. and J.L.M. acknowledge support from the Spanish Ministry and FEDER funds for the project AGL2017-85755-R, the I+D+i project PID2020-114942RB-I00 funded by MCIN/AEI/10.13039/ 501100011033, the i-LINK+2018 (LINKA20069) from CSIC, as well as funds from "Fundación Séneca" from Murcia Province (19896/GERM/15). E.M.-J. was supported by an Experienced Researcher Fellowship of the Humboldt Foundation. E.M-J. and C.P. acknowledge support from the Spanish Ministry of Science and Innovation (PID2020-116578RB-I00). F.A. is supported by ANID FONDECYT 1220358. H-W.H. and J-Z.H. are supported by the project (DP210100332) from Australian Research Council. S.A. is funded by ANID FONDECYT 1170995 and ANID ANILLO ACT192027. MB is supported by a Ramón y Cajal grant from Spanish Ministry of Science (RYC2021-031797-I). The contribution of TG and TUN was supported by the Research Program in Forest Biology, Ecology and Technology (P4-0107) and project V4-3098 of the Slovenian Research Agency. T.P.M. would like to acknowledge contributions from the National Research Foundation of South Africa and cities involved in the South African survey. J.D. and A. Rey acknowledge support from the FCT (IF/00950/2014 and SFRH/BDP/108913/2015, respectively). JPV is thankful to SERB (EEQ/2021/001083) and DST (DST/INT/SL/P-31/2021) and BHU-IoE (6031)-incentive grant for research and development. MCR acknowledges support from an ERC Advanced Grant (694368). AM acknowledged financial support from the PMRF, Ministry of Education - Government of India, India.

## Author contributions
M.D-B. and Y-R.L. developed the original idea of the analyses presented in the manuscript. M.D.-B. designed the field study and wrote the grant that funded the work. Field data were collected by M. D-B., D.J.E., Y-R.L., M.B., J.L.B-P., C.P., F.B., S.A., F.A., A.R.B., M.A.M-M., G.F.P-B., A.d.R., J.D.,T.G., J.G.I., A.M., T.P.M., T.U.N., A.Rey., A.L.T., A.Ro., C.T-D., P.T., L.W., J.W., E.Z., X.Z., J.P.V. and X-Q.Z. Lab analyses were done by M.D-B., Y-R.L., J.R., F.B., H-W.H., J-Z.H., J.L.M., P.T., N.C.C and C.S-L. Bioinformatic analyses were done by Y-R.L. and H-W.H. Statistical analyses were done by Y-R.L., M.D-B., X.M.Z. and X-Q.Z. The manuscript was written by Y-R.L. and M.D-B., and edited by M.G.A.v.d.H., D.J.E, E.M.J, E.Z., F.B., C.S., J.P.V., Q.H., W.T., M.C.R., Y-G.Z. and C.S-L. with contributions from all co-authors.

## Competing interests
The authors declare no competing interests.

## Additional information

[1]State Key Laboratory of Agricultural Microbiology, Huazhong Agricultural University, Wuhan 430070, China. [2]College of Resources and Environment, Huazhong Agricultural University, Wuhan 430070, China. [3]Plant-Soil Interactions, Agroscope, Zürich, Switzerland. [4]Department of Plant and Microbial Biology, University of Zürich, Zürich, Switzerland. [5]Multidisciplinary Institute for Environmental Studies (MIES), University of Alicante, P.O. Box 99, Alicante E-03080, Spain. [6]Department of Ecology, University of Alicante, PO Box 99, Alicante E-03080, Spain. [7]Centre for Ecosystem Science, School of Biological, Earth and Environmental Sciences, University of New South Wales, Sydney, NSW 2052, Australia. [8]CEBAS-CSIC. Department of Soil and Water Conservation. Campus Universitario de Espinardo, 30100 Murcia, Spain. [9]Department of Agricultural and Food Chemistry, Faculty of Sciences, Universidad Autónoma de Madrid, 28049 Madrid, Spain. [10]Institute of Biology, Freie Universität Berlin, Berlin 14195, Germany. [11]Berlin-Brandenburg Institute of Advanced Biodiversity Research (BBIB), Berlin 14195, Germany. [12]Faculty of Science, The University of Melbourne, Parkville 3010 VIC, Australia. [13]GEMA Center for Genomics, Ecology & Environment, Universidad Mayor, Santiago, Chile. [14]Instituto de Ecología y Biodiversidad (IEB), Santiago 7800003 CP, Chile. [15]Natural History Museum (Botany Unit), Obafemi Awolowo University, Ile-Ife, Nigeria. [16] Departamento de Biodiversidad, Ecología y Evolución, Facultad de Biología, Universidad Complutense de Madrid, C/Jose Antonio Novais 12, Madrid 28040, Spain. [17]INRAE, UR4 (URP3F),

Centre Nouvelle-Aquitaine-Poitiers, Lusignan 86600, France. [18]Museo Nacional de Ciencias Naturales, Consejo Superior de Investigaciones Científicas, Serrano 115 bis, 28006 Madrid, Spain. [19]Misión Biológica de Galicia, Consejo Superior de Investigaciones Científicas, Pontevedra, Spain. [20]Centre for Functional Ecology, Department of Life Sciences, University of Coimbra, 3000-456 Coimbra, Portugal. [21]Department of Forest Physiology and Genetics, Slovenian Forestry Institute, Ljubljana, Slovenia. [22]Department of Entomology, Washington State University, Pullman, WA 99164 USA, USA. [23]Department of Biochemistry, Genetics and Microbiology, DSI/NRF SARChI Chair in Marine Microbiomics, University of Pretoria, Pretoria 0028, South Africa. [24]Centre for Integrative Ecology, ICB, Universidad de Talca, Talca, Chile. [25]CEAZA, Universidad Católica del Norte, Coquimbo, Chile. [26]Laboratório de Sistemática Vegetal, Departamento de Botânica, Instituto de Ciências Biológicas, Universidade Federal de Minas Gerais, Av. Antônio Carlos, 6627, Pampulha, Belo Horizonte 31270-901 MG, Brazil. [27]Instituto de Ciencias Agrarias, Consejo Superior de Investigaciones Científicas, Serrano 115 bis, 28006 Madrid, Spain. [28]Instituto de Geología, Universidad Nacional Autónoma de México, Ciudad Universitaria, México D.F 04510 CP, México. [29]Departamento de Botânica e Ecologia, Instituto de Biociências, Universidade Federal de Mato Grosso, Av. Fernando Corrêa, 2367, Boa Esperança, Cuiabá 78060-900 MT, Brazil. [30]Microbiome Network and Department of Agricultural Biology, Colorado State University, Fort Collins 80523 CO, USA. [31]Grupo de Biodiversidad y Cambio Global (BCG), Departamento de Ciencias. Básicas, Universidad del Bío-Bío, Campus Fernando May, Chillán, Chile. [32]Plant-Microbes Interaction Lab, Institute of Environment and Sustainable Development, Banaras Hindu University, Varanasi 221005 Uttar Pradesh, India. [33]Institute of Grassland Science/School of Life Science, Northeast Normal University, and Key Laboratory of Vegetation Ecology, Ministry of Education, Changchun, 130024 Jilin, China. [34]Department of Natural Resources, Agricultural Research Organization, Institute of Plant Sciences, Gilat Research Center, Negev 8531100, Israel. [35]State Key Laboratory of Desert and Oasis Ecology, Xinjiang Institute of Ecology and Geography, Chinese Academy of Sciences, Urumqi, China. [36]State Environmental Protection Key Laboratory of Soil Health and Green Remediation, Wuhan 430000, China. [37]State Key Laboratory of Urban and Regional Ecology, Research Center for Eco-Environmental Sciences, Chinese Academy of Sciences, Beijing 100085, China. [38]Laboratorio de Biodiversidad y Funcionamiento Ecosistémico. Instituto de Recursos Naturales y Agrobiología de Sevilla (IRNAS), CSIC, Av. Reina Mercedes 10, Sevilla E-41012, Spain. [39]Unidad Asociada CSIC-UPO (BioFun)., Universidad Pablo de Olavide, Sevilla 41013, Spain. ✉e-mail: yrliu@mail.hzau.edu.cn; m.delgadobaquerizo@gmail.com

