## [Peer Review File · Nature Communications]

Soil contamination in nearby natural areas mirrors that in urban greenspaces worldwideReviewer 1 (Remarks to the Author):

Reviewer report

The submitted manuscript represents an ambitious attempt to quantify and explain soil contamination levels in 'urban greenspaces' and contrasting, yet comparable, 'natural ecosystems' on a global scale based on 56 paired ecosystems (i.e. 56 urban greenspace soils and 56 natural or semi-natural ecosystem soils). Three remote maritime ecosystems from Antarctica were also included in the survey as reference soils. Soil contamination is evaluated using several lines of evidence; i.e. metal(loid)s, pesticides, microplastics, antibiotic resistance genes, and other functional genes belonging to the soil microbiome. At its core the study represents a survey with no firm hypotheses for the study, and multivariate statistical approaches (e.g. structural equation modelling) is used to explain soil contamination levels using both socio-economic and environmental factors as model elements. According to the abstract the main conclusions are [quotes]:

- 1) urban greenspaces and adjacent natural ecosystems shared similar levels of multiple soil contaminants (metal(loid)s, pesticides, microplastics and antibiotic resistance genes) across the globe.
- 2) Human influence explained many forms of soil contamination worldwide.
- 3) Socio-economic factors were integral to explaining the occurrence of soil contaminants worldwide, with population density being positively associated with contents of metal(loid)s and microplastics.
- 4) Increased levels of multiple soil contaminants were linked with changes in microbial traits including genes associated with environmental stress resistance, nutrient cycling, and pathogenesis.

The authors go on to conclude that [quotes]:

- 5) Human-driven soil contamination in nearby natural areas mirrors that in urban greenspaces globally.
- 6) Soil contaminants have the potential to cause important biological stress responses, with dire consequences for ecosystem sustainability and human wellbeing.

The survey part (Point 1) has value as it is comprehensive and based on within-study-standardized methodologies. Nevertheless, I note that not all soils were analyzed for all soil contamination parameters and that the number of within-site replicates were also not standardized for all parameters. More importantly, the remaining five points (Points 2-6 above) have limited significance as they are largely based on multivariate statistical approaches that are not sufficiently justified by the authors (see below). Hence, there is for instance no rationale provided for how gross national product data or other socio-economic parameters can explain soil contamination levels. Correlations do not prove causality and the value of this part of the work is thus limited. This conclusion is further substantiated by the fact that some of the conclusions are borderline trivial in that the submitted manuscript 'suggests' findings that are already established knowledge.

Soil contaminants were wisely chosen and represents several contrasting lines of evidence and the global survey using within-study-standardized, robust methodologies has high value. Nevertheless, the study has significant weaknesses. What is a 'natural ecosystem' and which criteria were used for their selection? All ecosystems in and around cities will obviously be impacted by human activities. The authors sometimes use the term "semi-natural" just adding to the confusion. There is no detailed descriptions of all chosen ecosystems and it is thus impossible to review the selection of ecosystems. The authors also do not provide a robust scientific justification of the a priori conceptual model used for structural equation modelling. This is especially the case for the socio-economic factors. In general, the level of detail in the methods description does not allow for the work to be reproduced.

Specific comments:

Line 129: I would prefer not to use the word 'diffusion' for transport of contaminants

across large spatial scales.

Line 181: I do not think that it makes sense to write 'natural vs. anthropogenic' as all ecosystems near urban centers will be affected by anthropogenic activities.

Line 189: The meaning of the phrase 'mean baseline levels' must be properly explained. I question the validity of this phrase as all studied ecosystems will already have been impacted from anthropogenic activities for many decades at the time of sampling. Hence, the measurements do not provide a 'baseline', but rather just a snapshot of soil contamination levels at the time of sampling.

Line 191-192: Why do you write 'strikingly', if it has already been shown earlier in local reports?

Lines 198-200: This is a very bold statement for multiple reasons: 1) just based on statistical associations (not necessarily on causal relationships); 2) stress resistance is a very vague ill-defined term and 3) not clear in the manuscript how pathogenesis traits are defined.

Lines 203-205: This is not novel. The results 'suggest' something that we already know.

Lines 209-211: Please clarify this sentence to justify the contributions from the submitted study as opposed to the cited reference.

Lines 216-218: This sentence about possible environmental regulation of Hg seems a little off-topic.

Line 232: Be more specific about pesticides covered (and not covered) by the analytical techniques used.

Lines 279-286: It is not possible to discuss differences in ARG diversity without considering detection limits for ARG detection; ARGs may be there in copy numbers below the detection limit.

Line 291: Hardly a surprise that we find plastic where there are people.

Lines 298-299: Again hardly a surprise. Scientific value of this statement is very low.

Lines 302-304: The authors cannot infer causal relationships, but of course the statement is true. Something that we know in general, but we do not need structural equation modelling exercises to reach this conclusion!

Line 332: 'wide range of genes' is too vague to give any firm meaning to the reader. The used approach is not reported in sufficient detail.

Line 368-372: I miss a rational for the selection of soils for this study and I miss detailed information about the sites (give precise geographical coordinates) and the degree to which they are impacted by anthropogenic activities. I also miss details on soil types and soil physicochemical characteristics.

Line 380: Please explain what 'human development index' is.

Lines 393-395: What logistic reasons? It is a major weakness and makes the study less comprehensive than it probably should have been to really inform on global scale soil contamination levels. A total of 54 soil samples is simply not enough in my opinion (see also lines 438-442).

Line 462: Seems inconsistent with previous information (please check).

Line 489: I am not an expert on microplastic analysis, but it seems problematic to me to keep soils in plastic bags before they are being analyzed.

Line 501: Seems inconsistent with text in lines 438-442.

Lines 516-517: Selection criteria?

Lines 537-539: Please provide full details on these genes; I am critical on this procedure.

Lines 585-586: There is no justification of the a priori conceptual model used for SEM.

Line 620: Selection of sites is critical for a study like this. Did these two persons select all the soils and based on what criteria?

Figure 1: Very hard to read/understand; please explain 'response ratio' and 'color cycles'. In my view it does not make much sense to plot the data on top of a world map as the spatial resolution of data is far too low to give any meaning.

Figure 3 looks nice, but it does not really help explain the data and the figure is not used much in the paper. I would say that it is redundant.

Reviewer 2 (Remarks to the Author):

Key results

The authors have completed a large scale study to assess various soil contaminants in both urban and rural/natural areas around the world. The global study allows for a great overview of the effects of the contaminants in soil. This study can create a good foundation for future studies, when assessing risk of soil contaminants on health.

Validity

This is a large scale study with collaborations across the globe. The methods are explained in detail. The authors were able to show correlations between urban and natural areas due to human derived soil contamination.

Significance

The study has good potential to increase knowledge on soil contamination and health. This study shows a nice overview of soil contamination across the globe in both urban and natural areas, showing implications for human and environmental health. This adds novel information to the literature in this area. This is an interesting manuscript.

Data and methodology

The methods are valid and reproducible. The methods used align with the conclusions and and implications made in the manuscript.

Analytical approach

The statistical analysis is sound and meet the expected standards in this field. The analysis used lead to the interpretations and conclusions made in the manuscript.

Suggested improvements

It may be helpful for the reader to use subtitles in you results and discussion for a clear flow the reader can follow

In line 218 - you may want to add reasoning for your use of the Finnish Guidelines.

In line 393- you may want to add in information on your "logistic reasons"

In figure 2-you may want to add in information regarding the arrows for the SEM (a description for red vs blue vs gray)

Clarity and context

The manuscript is well written and clear. The results have sufficient context in this study and compare to literature.

References

The references used are necessary and accurate.

Reviewer 3

Review report

Manuscript: Soil contamination in nearby natural areas mirrors that in urban greenspaces worldwide

Authors: Yu-Rong Liu et al.

General comments:

The manuscript compares the contaminants in urban soils with those presents in adjacent natural soils to define the environmental factors associated with the multiple contaminations. The work can be considered a relevant contribution in the field.

The document is well written and indicates straightforwardly the main findings of the results. The study includes a huge number of soils from a wide range of environmental and climatic conditions with a wide geographical distribution from 17 countries around the world. The data are profusely evaluated statistically and schematized in well-designed but complicated figures.

Throughout the manuscript there are many errors in the grammar/syntax. Most are only

small, but some could lead to misinterpretation of the sentence. For example, in line 605, "new hypothesis" needs to be changed to "new hypotheses" or "a new hypothesis", according to what the authors want to say. In particular, there are numerous errors in lines 466-483. As the manuscript is written in American English, lists should be written as (for example) "A, B, and C" but in many cases the final comma is missing. "Phosphorus" should be used throughout the text.

One of the authors, whose mother tongue is English, should perform a careful linguistic review of the manuscript (starting on line 158; before that, the text is acceptable), including the text in the supplementary data.

My main concern is related to the urban soils. Such soils from city gardens and parks are profusely altered by different actions, such as construction works, re-design, removal or addition of top soils, application of different amendments (such as compost), etc. which alter their characteristics in such way that they no longer reflect the parent material. This can make the comparison with the natural soil difficult or unrealistic.

Discussion concerning the influence of the wind on contaminant dispersion is lacking. For example, Lines 260-263, the authors should clarify the likely sources of the microplastics at the Antarctica sites ("interior areas" is a vague description). What are the most likely places from which and the mechanisms by which (air, sea) they arrived at these sites? I imagine that this will be related to the directions of sea currents and/or prevailing winds. As there is human activity in Antarctica (research stations etc.), could this contribute?

Specific comments.

Line 154. I think "controlling for" should be replaced by "allowing for" or "taking into account".

Line 170. Change to "in certain soils". Only specific types of soil (ultramafic etc.) are naturally enriched in such elements.

Line 173. I think the first "density" should be removed.

Lines 272-273 or 280-282. Add information on the mechanism of transfer of ARGs to urban greenspaces. Also, are antibiotics used for veterinary purposes in urban areas a factor?

Lines 306-307. Clarify this sentence: is one source of the ARGs the animal excreta used to produce the organic fertilizers?

The material and methods section needs improving. The soil type and/or classification for natural ecosystems should be indicated to show the variety of soils used.

Line 377. The sentence "Agricultural fields were not included in this study" is repeated in line 381.

Line 382. Change "nearby" to "near"

Subsection Environmental factors (from line 403):

Lines 407-412. The description of the general methods for soil analysis should be moved to the following subsection (renamed as Soil analysis). Also, include a new heading for general soil analysis.

The number of soil samples for each contaminant analysis is not clear, as different numbers are indicated in each subsection of the specific contaminant (see lines 422-423, 438-442, 462-463, 501 and 516). The number of soil samples for each contaminant should be clearly indicated in the subsection Field survey and soil sampling (lines 383-390). Avoid repetition in the respective subsection for each contaminant type.

Line 425. Is "trace mixed acids" a mistake?

Line 427. Change "using an" to "by"

Line 430. I think "aqua regia" should be written in italics.

Line 441. Change MP to microplastics

Lines 463-4. State what "5 mm" refers to: diameter, length of the longest side?

Line 468: change retain to maintain

Line 475. Should it be "flotation" instead of "floatation"?

Check for typing mistakes (see line 479).

Line 493. "carefully" seem to be in the wrong place.

Lines 518-519. Remove sentence to avoid repetitions in the experimental procedure.

Line 546. Remove ")"

Line 550. Change "mean" to "indicate" or "represent" ("mean" is confusing here as it is also a statistical term).

Line 593. Remove the first "RMSEA".

Figure 1. The sentence on lines 800-803 seems to be incorrectly written.

Figure 2. This figure is not clear. What do the lines and arrows in blue and red mean in panel A? They are not yellow boxes, but orange. Some abbreviations are not specified.

Responses to reviewer's comments

Reviewer 1 (Remarks to the Author):

The submitted manuscript represents an ambitious attempt to quantify and explain soil contamination levels in 'urban greenspaces' and contrasting, yet comparable, 'natural ecosystems' on a global scale based on 56 paired ecosystems (i.e. 56 urban greenspace soils and 56 natural or semi-natural ecosystem soils). Three remote maritime ecosystems from Antarctica were also included in the survey as reference soils. Soil contamination is evaluated using several lines of evidence; i.e. metal(loid)s, pesticides, microplastics, antibiotic resistance genes, and other functional genes belonging to the soil microbiome. At its core the study represents a survey with no firm hypotheses for the study, and multivariate statistical approaches (e.g. structural equation modelling) is used to explain soil contamination levels using both socio-economic and environmental factors as model elements. According to the abstract the main conclusions are [quotes]:

- 1) urban greenspaces and adjacent natural ecosystems shared similar levels of multiple soil contaminants (metal(loid)s, pesticides, microplastics and antibiotic resistance genes) across the globe.
- 2) Human influence explained many forms of soil contamination worldwide.
- 3) Socio-economic factors were integral to explaining the occurrence of soil contaminants worldwide, with population density being positively associated with contents of metal(loid)s and microplastics.
- 4) Increased levels of multiple soil contaminants were linked with changes in microbial traits including genes associated with environmental stress resistance, nutrient cycling, and pathogenesis.

The authors go on to conclude that [quotes]:

- 5) Human-driven soil contamination in nearby natural areas mirrors that in urban greenspaces globally.
- 6) Soil contaminants have the potential to cause important biological stress responses, with dire consequences for ecosystem sustainability and human wellbeing.

The survey part (Point 1) has value as it is comprehensive and based on within-study-standardized methodologies.

1. Thank you for the positive and constructive comments on our manuscript. We appreciate that the reviewer highlighted the main findings and the importance of our work. We aimed at addressing all your comments below.

Nevertheless, I note that not all soils were analyzed for all soil contamination parameters and that the number of within-site replicates were also not standardized for all parameters.

2. We appreciate this comment. We acknowledge the potential limitations of an unbalanced number of samples for the analyses of the four categories of soil contaminants. To the best of our knowledge, however, this is the largest paired global survey that aimed at comparing soil contamination in urban greenspaces vs. nearby reference “natural” areas. As this reviewer can imagine, the logistics of organizing a global field sampling of paired urban and natural areas are not easy. Our study provides a first attempt to quantify multiple soil contaminants across a worldwide spatial distribution. Even so, due to sample availability and resource limitation, we did not analyze all soils for each category of contaminants (i.e., metal(loid)s, pesticides, microplastics and ARGs). Many of the analyses included in this study (e.g., polymer identification of microplastics and 46 pesticide residues) are highly costly and time-consuming, which require a relatively large amount of sample stored following very specific conditions (see Methods). As explained in lines 411-429, we used a subset of samples for contaminant analyses.

We have clarified these important points and acknowledge the limitations of our design in the revised manuscript (lines 392-395; lines 427-429). In particular, we stated that “Due to sample availability and resource limitation, we did not analyze all soils for each category of contaminant. Many of the analyses included in this study (e.g., polymer identification of microplastics and 46 pesticide residues) are highly costly and time-consuming, which require a relatively large amount of sample for contaminant extraction (see Methods below). We thus focused on the subset of sites potentially impacted by specific contaminants according to advice from local greenspace managers and environmental scientists. Therefore, we attempted to cover the entire gradient of conditions by focusing on a subset of the samples. Thus, we measured (metal(loid)s and ARGs in all the 336 composite samples from 112 plots, microplastics in 64 composite samples from the selected 64 plots and pesticide residues in 54 composite samples from the selected 54 plots (Supplementary Fig. 1, Table 2). For comparability, we standardized within-site replicates for metal(loid)s and ARGs (e.g., 112 plots \times 3 composite samples = 336 samples). In all cases, the subsets of samples were selected to cover the entire biogeographic range along with a broad range of environmental gradients (Supplementary Fig. 1 and Table 1). To the best of our knowledge, this is the largest paired global survey that aimed at comparing soil contamination in urban greenspaces vs. nearby reference “natural” areas. We acknowledge, however, the potential limitations of an unbalanced number of samples for the analyses of the four categories of soil contaminants”.

Correlations do not prove causality and the value of this part of the work is thus limited. This conclusion is further substantiated by the fact that some of the conclusions are borderline trivial in that the submitted manuscript 'suggests' findings that are already established knowledge.

More importantly, the remaining five points (Points 2-6 above) have limited significance as they are largely based on multivariate statistical approaches that are not sufficiently justified by the authors (see below). Hence, there is for instance no

rationale provided for how gross national product data or other socio-economic parameters can explain soil contamination levels.

3. We agree that correlation analyses cannot prove causality, and it was not our intention to imply this. We have now toned down our language to avoid any confusion in this direction. However, we think that the approach used is still valuable and is, as reviewer 2 highlighted, “sound and meet the expected standards in this field”. We have now clarified that:

- We acknowledge that multivariate statistical approaches have limitations in explaining the distribution and ecological consequences of soil contaminants; however, it would not be feasible to conduct an experimental study considering multiple soil contaminants simultaneously at a global scale. Observational studies are fundamental for developing new hypotheses which can then be tested in the laboratory and the field under controlled conditions as done in many studies (Crowther et al., 2019, Science; Peters et al., 2019, Nature) (lines 367-368). The combined statistical methods have been widely used for large-scale studies, disentangling complex relationships between variables (Bowker et al., 2005; Jing et al., 2015; Laughlin et al., 2007; Semchenko et al., 2018; Maestre, et al. 2022). We believe that future experimental studies testing further links between specific soil contaminations and microbial functional traits would advance our understanding. We have now clarified this point and acknowledged the limitations (see lines 318-320; lines 351-355).

- Our observational study is essential for understanding the fate and consequences of soil contamination, especially for paired urban and adjacent natural areas at a global scale. This study considers multiple environmental and socio-economic factors which are highly challenging to control under experimental conditions at such a large scale. The main findings of this study are based on block design, providing a systematic and multidimensional understanding of soil contaminants in paired urban and natural areas worldwide. The multivariate analyses help us infer associations between factors while allowing for multiple factors simultaneously. Thus, these analyses are fundamental and fully aligned with the aim of this study: a global and multidimensional assessment of soil contamination in urban greenspaces. Such knowledge has been not previously reported. It is also essential to quantify how contaminated natural areas surrounding cities are, compared with highly managed urban greenspaces (lines 177-182).

- Finally, we have now provided the rationale of our prior model in Methods and a new Fig. S18. For example, countries with a high gross national product generally invest more to waste management and have more restrictive policies in terms of environment protection, than countries with low gross national product (lines 602-607).

In summary, through these analyses, we build a strong body of evidence suggesting strong and intuitive relationships among our study attributes. We have been careful to avoid terms that might imply causality. Instead, we have now

explained the significance of our work in the revised manuscript (lines 201-203; lines 367-368). While earlier studies focused on the global distribution of a wide range of soil biota (e.g., Delgado-Baquerizo et al., 2018, *Science*; Guerra et al., 2022, *Nature*), a global evaluation of the presence of a range of important soil contaminants in natural and neighboring urban areas is missing so far.

Soil contaminants were wisely chosen and represents several contrasting lines of evidence and the global survey using within-study-standardized, robust methodologies has high value.

4. Thanks.

Nevertheless, the study has significant weaknesses. What is a 'natural ecosystem' and which criteria were used for their selection? All ecosystems in and around cities will obviously be impacted by human activities. The authors sometimes use the term “semi-natural” just adding to the confusion. There is no detailed descriptions of all chosen ecosystems and it is thus impossible to review the selection of ecosystems.

5. Thanks for this pertinent comment. We agree that all ecosystems in and around cities can be impacted by human activities. We have now included a more detailed definition of the natural ecosystem to avoid further confusion. We defined natural areas as natural and semi-natural ecosystems not subjected to human management near urban greenspaces. Natural areas are about 20 km from the urban greenspaces, which include forests, grasslands, shrublands, or relict forests with their original vegetation. We have clarified this in lines 375-380.

The authors also do not provide a robust scientific justification of the a priori conceptual model used for structural equation modelling. This is especially the case for the socio-economic factors. In general, the level of detail in the methods description does not allow for the work to be reproduced.

6. We agree that this is fundamental. We have now provided a detailed justification of our prior model in new Fig. S18, explaining all the paths in our models supported by current knowledge.

Specific comments:

Line 129: I would prefer not to use the word 'diffusion' for transport of contaminants across large spatial scales.

7. We have now used “dispersion” instead of “diffusion”.

Line 181: I do not think that it makes sense to write 'natural vs. anthropogenic' as all ecosystems near urban centers will be affected by anthropogenic activities.

8. We agree with this point. As explained in response #5, we have now clarified that we refer to natural areas defined as natural and semi-natural ecosystems not subjected to human managements near urban greenspaces.

Line 189: The meaning of the phrase 'mean baseline levels' must be properly

explained. I question the validity of this phrase as all studied ecosystems will already have been impacted from anthropogenic activities for many decades at the time of sampling. Hence, the measurements do not provide a 'baseline', but rather just a snapshot of soil contamination levels at the time of sampling.

9. Thanks for your constructive comments. Considering the potential impacts of human activities on these natural areas, we have now rewritten it as “Our data further provide a snapshot of multiple soil contaminants in natural areas, comparing with adjacent urban greenspace worldwide” (lines 185-186).

Line 191-192: Why do you write 'strikingly', if it has already been shown earlier in local reports?

10. This word has been deleted.

Lines 198-200: This is a very bold statement for multiple reasons: 1) just based on statistical associations (not necessarily on causal relationships); 2) stress resistance is a very vague ill-defined term and 3) not clear in the manuscript how pathogenesis traits are defined.

11. We have rewritten this sentence based on the results from our observations and analyses as “Finally, we found that levels of soil contaminants were significantly correlated with the relative abundance of important functional genes associated with stress resistance, nutrient cycling, pathogenicity and microbial metabolism (Fig. 3)”. We also explained the meaning of stress resistance and pathogenesis traits in the text and Methods and an additional table (Table S9), “The resistance traits included the resistance genes to stress of metal(loid)s, drugs and pathogens. Detailed information on microbial traits were assessed by the gene abundance in the soil microbiome is shown in Supplementary Table 9” (lines 556-560).

Lines 203-205: This is not novel. The results 'suggest' something that we already know.

12. We understand the reviewer’s concern. In this study, we focused on multiple dimensions of soil contamination simultaneously, by comparing paired urban and adjacent natural areas. To the best of our knowledge, this has not been attempted previously. We have now conducted additional analyses of contaminant number effects on the microbiome, and found that an increasing number of soil contaminants can have a greater effect on the proportion of the microbial functional traits (see Supplementary Method, new Fig. S6), suggesting the importance of considering multiple soil contaminants simultaneously. In particular, we found greater gene proportions of multidrug resistance and *Listeria prophages* in the soils with higher levels of multi-contamination. We have clarified these important points in the revised manuscript. “Importantly, we found that an increasing number of soil contaminants can have a greater effect on the proportion of the microbial traits (Supplementary Fig. 6)”; “The results revealed that adequate conservation and functioning of terrestrial ecosystems requires an assessment of risk associated with multiple dimensions of soil contamination”

(lines 201-203).

Lines 209-211: Please clarify this sentence to justify the contributions from the submitted study as opposed to the cited reference.

13. The cited reference showed that human activities had different effects on the accumulation of metal(loid)s in urban soils from a local study (Xiamen city in China). Our study is somewhat different, showing a pattern of soil metal(loid) accumulation in urban greenspaces compared with those in adjacent natural areas worldwide. We now state “This different accumulation of metal(loid)s in urban greenspaces compared with natural areas could be related to geographic variations in the extent of metal(loid) emissions from human activities, though the exact mechanisms remain unknown”.

Lines 216-218: This sentence about possible environmental regulation of Hg seems a little off-topic.

14. The sentence has been removed.

Line 232: Be more specific about pesticides covered (and not covered) by the analytical techniques used.

15. We thank the reviewer for this remark. We have now added specific examples of the pesticides covered and not covered within our analytical techniques (lines 230-232; lines 481-483).

Lines 279-286: It is not possible to discuss differences in ARG diversity without considering detection limits for ARG detection; ARGs may be there in copy numbers below the detection limit.

16. Thanks for this important point. We have now explained the limitation of detecting soil ARGs (lines 288-289).

Line 291: Hardly a surprise that we find plastic where there are people.

17. We agree that microplastics are known to derive from human activity, but the extent differs across ecosystems and regions. Additionally, they can be transported and accumulated in remote areas. Microplastics in soil are related to multiple factors such as regional economic development, population density, and soil management, so here we showed the relative importance of these factors. We revised this as “Structural equation modelling (SEM) and linear mixed-effects models consistently revealed that regional population density was the most important socio-economic factor associated with microplastics” (lines 296-298).

Lines 298-299: Again hardly a surprise. Scientific value of this statement is very low.

18. We have removed the sentence.

Lines 302-304: The authors cannot infer causal relationships, but of course the statement is true. Something that we know in general, but we do not need structural equation modelling exercises to reach this conclusion!

19. As explained in response #3, we cannot infer causality. Instead, we focused on the relative importance when accounting for multiple human and natural factors on soil contamination by structural equation modelling. We have revised this sentence as “According to our modelling analyses, human factors such as soil fertilization played more important roles than natural factors in influencing the abundance of pesticides and ARGs worldwide” (lines 308-310).

Line 332: 'wide range of genes' is too vague to give any firm meaning to the reader. The used approach is not reported in sufficient detail.

20. We have now removed these repeated statements to avoid confusion and provided detailed information on the used approach (see lines 556-560 and added Table S9).

Line 368-372: I miss a rationale for the selection of soils for this study and I miss detailed information about the sites (give precise geographical coordinates) and the degree to which they are impacted by anthropogenic activities. I also miss details on soil types and soil physicochemical characteristics.

21. The paired urban and natural areas were selected across a broad range of environmental gradients worldwide, and also considered the soil contamination attributes and sampling feasibility. We have now added information on the precise geographical coordinates. Considering potential impacts of human activities on the natural and semi-natural ecosystems without human management, we defined them as “natural areas”. We have also provided details on the sampling rationale, soil types and physicochemical characteristics, as well as the levels of soil contaminants in the paired ecosystems (see lines 375-384; lines 411-429; Table S1 and Table S5).

Line 380: Please explain what 'human development index' is.

22. We have explained this in the revised manuscript (lines 390-391).

Lines 393-395: What logistic reasons? It is a major weakness and makes the study less comprehensive than it probably should have been to really inform on global scale soil contamination levels. A total of 54 soil samples is simply not enough in my opinion (see also lines 438-442).

23. We appreciate the comment. As explained in response # 2, we did not analyze all soils for the four categories of contaminants (341 individual soil contaminants in total) due to sample availability and resource limitation. Many of the analyses included in this study (e.g., polymer identification of microplastics and 46 pesticide residues) are extremely costly and also require a relatively large amount of sample for contaminant extraction (see Methods). In this respect, we tried our best to focus on the subset of sites potentially impacted by certain contaminants according to local greenspace managers and environmental scientists, while covering the entire gradient of conditions. We have used the term it as “worldwide” instead of “global” in the revised manuscript. The sentence in line 441-442 was wrong, and has been removed.

Line 462: Seems inconsistent with previous information (please check).

24. We analyzed microplastics in 64 composite soil samples (from 32 paired urban and natural areas). We have corrected this point.

Line 489: I am not an expert on microplastic analysis, but it seems problematic to me to keep soils in plastic bags before they are being analyzed.

25. We understand the reviewer's concern. We determined the spectroscopic signature of all the bags used for the sampling, which did not match any of the signature of microplastics found in the samples. See lines 511-513.

Line 501: Seems inconsistent with text in lines 438-442.

26. We have corrected the typo in lines 441-442. We indeed analyzed ARGs in all the soil of 112 plots (336 composite soils).

Lines 516-517: Selection criteria?

27. Metagenomics analyses are expensive, and we thus did our best to cover the entire gradient of conditions focusing on a subset of the samples as regularly done in many other studies (Howe et al., 2014; Bahram et al., 2018; Delgado-Baquerizo et al., 2021). Please see Fig. S1C and Table S1 for detailed information on these sampling sites.

Lines 537-539: Please provide full details on these genes; I am critical on this procedure.

28. That is an important point, and we have now provided details on these genes in the revised manuscript (see Table S9; lines 556-560). The resistance traits included the genes mediating resistance to the stress of metal(loid)s, drugs and pathogens.

29. Lines 585-586: There is no justification of the a priori conceptual model used for SEM.

30. We have now provided a justification of the priori conceptual model used for SEM (see lines 598-603 and new Fig. S18).

Line 620: Selection of sites is critical for a study like this. Did these two persons select all the soils and based on what criteria?

31. Sorry for the misleading statement. The two persons assisted during sampling, while our global co-authors selected the study sites, and many co-authors were responsible for the soil sampling in the fields. We revised it as "We are grateful for the assistance of Yunyun Hao and Xuemei Han during soil sampling".

Figure 1: Very hard to read/understand; please explain 'response ratio' and 'color cycles'. In my view it does not make much sense to plot the data on top of a world map as the spatial resolution of data is far too low to give any meaning.

32. We appreciate your constructive comments. We have now presented the data as a new Fig. 1 to avoid confusion, and also provided detailed information on the sites

(see Table S1).

Figure 3 looks nice, but it does not really help explain the data and the figure is not used much in the paper. I would say that it is redundant.

33. Thanks. We thus removed this Figure according to your suggestion.

Reviewer 2 (Remarks to the Author):

Key results

The authors have completed a large scale study to assess various soil contaminants in both urban and rural/natural areas around the world. The global study allows for a great overview of the effects of the contaminants in soil. This study can create a good foundation for future studies, when assessing risk of soil contaminants on health.

34. Thank you for the positive and constructive comments. We appreciated that this reviewer highlighted the importance of our work.

Validity

This is a large scale study with collaborations across the globe. The methods are explained in detail. The authors were able to show correlations between urban and natural areas due to human derived soil contamination.

35. Thanks.

Significance

The study has good potential to increase knowledge on soil contamination and health. This study shows a nice overview of soil contamination across the globe in both urban and natural areas, showing implications for human and environmental health. This adds novel information to the literature in this area. This is an interesting manuscript.

36. Thanks for highlighting the significance of our study.

Data and methodology

The methods are valid and reproducible. The methods used align with the conclusions and and implications made in the manuscript.

37. Thanks for the positive comments on our methodology and data.

Analytical approach

The statistical analysis is sound and meet the expected standards in this field. The analysis used lead to the interpretations and conclusions made in the manuscript.

38. Thanks for the positive comments on the analytical approaches used in this study.

Suggested improvements

It may be helpful for the reader to use subtitles in you results and discussion for a

clear flow the reader can follow

39. Thanks for the suggestion. We have now added subtitles in the revised manuscript.

In line 218 - you may want to add reasoning for your use of the Finnish Guidelines.

40. Added. The standard has been widely used in similar approaches to assess soil contamination.

In line 393- you may want to add in information on your "logistic reasons"

41. We have explained the logistic reasons as "Due to sample availability and resource limitation...." (lines 411-417).

In figure 2-you may want to add in information regarding the arrows for the SEM (a description for red vs blue vs gray)

42. We have now updated this figure for clarity and consistently used black lines to avoid confusion.

Clarity and context

The manuscript is well written and clear. The results have sufficient context in this study and compare to literature.

43. Thanks again for the positive comments on our manuscript.

References

The references used are necessary and accurate.

44. Thanks!

Reviewer 3

Review report

Manuscript: Soil contamination in nearby natural areas mirrors that in urban greenspaces worldwide

Authors: Yu-Rong Liu et al.

General comments:

The manuscript compares the contaminants in urban soils with those presents in adjacent natural soils to define the environmental factors associated with the multiple contaminations. The work can be considered a relevant contribution in the field. The document is well written and indicates straightforwardly the main findings of the results. The study includes a huge number of soils from a wide range of environmental and climatic conditions with a wide geographical distribution from 17 countries around the world. The data are profusely evaluated statistically and schematized in well-designed but complicated figures.

45. Thank you for the positive and constructive comments and for highlighting the relevance of our work. We aimed at addressing your concerns below.

Throughout the manuscript there are many errors in the grammar/syntax. Most are only small, but some could lead to misinterpretation of the sentence. For example, in line 605, “new hypothesis” needs to be changed to “new hypotheses” or “a new hypothesis”, according to what the authors want to say. In particular, there are numerous errors in lines 466-483. As the manuscript is written in American English, lists should be written as (for example) “A, B, and C” but in many cases the final comma is missing. “Phosphorus” should be used throughout the text.

46. We have now corrected syntax errors pointed out by this reviewer and others throughout the manuscript, as well as those in the figure captions. Please see our revisions highlighted in the revised manuscript.

One of the authors, whose mother tongue is English, should perform a careful linguistic review of the manuscript (starting on line 158; before that, the text is acceptable), including the text in the supplementary data.

47. Prof. Eldridge, a native English speaker and coauthor of the paper, has reviewed the manuscript for grammar and expression.

My main concern is related to the urban soils. Such soils from city gardens and parks are profusely altered by different actions, such as construction works, re-design, removal or addition of top soils, application of different amendments (such as compost), etc. which alter their characteristics in such way that they no longer reflect the parent material. This can make the comparison with the natural soil difficult or unrealistic.

48. We understand the reviewer’s concerns. We agree that soils in urban ecosystems are thought to be quite different from those in natural areas. However, these are still soils, which follow similar rules in terms of biology and biogeochemistry (Li et al, 2018). All the selected urban greenspaces are well established and have existed for many decades. We do not say that these soils are similar to natural soils. Instead, we compared the contamination levels of urban greenspaces with those of natural areas. We found that these urban soils are more similar, in terms of contamination, to those in natural areas surrounding cities than previously thought. This is one of the major novelties in our work. We have now highlighted this in lines 378-384.

Discussion concerning the influence of the wind on contaminant dispersion is lacking. For example, Lines 260-263, the authors should clarify the likely sources of the microplastics at the Antarctica sites (“interior areas” is a vague description). What are the most likely places from which and the mechanisms by which (air, sea) they arrived at these sites? I imagine that this will be related to the directions of sea currents and/or prevailing winds. As there is human activity in Antarctica (research stations etc.), could this contribute?

49. Thanks for the constructive comments. We have now provided additional discussion on microplastic dispersion by air and sea, depending on the strength and direction of winds. Human activities (e.g., that from research stations and tourism) may also contribute to the accumulation of the microplastics in the soils of Antarctica sites. We have now revised the sentences “This could be related to microplastic dispersion from Antarctic research stations and other continents by sea and air, though this would depend on wind strength and direction” (lines 259-261). “Other activities such as tourism may also contribute to the accumulation of the microplastics in the soils of Antarctica sites” (lines 264-265).

Specific comments.

Line 154. I think “controlling for” should be replaced by “allowing for” or “taking into account”.

50. Revised as suggested.

Line 170. Change to “in certain soils”. Only specific types of soil (ultramafic etc.) are naturally enriched in such elements.

51. Changed.

Line 173. I think the first “density” should be removed.

52. Removed.

Lines 272-273 or 280-282. Add information on the mechanism of transfer of ARGs to urban greenspaces. Also, are antibiotics used for veterinary purposes in urban areas a factor?

53. Thanks for the constructive suggestion. We have now added information on the mechanism of transfer of ARGs to urban greenspaces. We also discussed the effects of antibiotics used for veterinary purposes on the ARGs in the urban greenspaces as “...linked to the widespread use of antibiotics in medical and agricultural industries (e.g., veterinary medicines). The intense presence of pet excreta, atmospheric deposition, and population movement may be the main transfer routes of ARGs to the soil of urban greenspaces” (lines 275-277).

Lines 306-307. Clarify this sentence: is one source of the ARGs the animal excreta used to produce the organic fertilizers?

54. Yes, animal manure is one of sources of the ARGs. We revised this (line 312).

The material and methods section needs improving. The soil type and/or classification for natural ecosystems should be indicated to show the variety of soils used.

55. We have now provided additional table (new Table S1) to show soil type information, and also improved the description of the other parts in Methods (e.g., lines 375-384; lines 411-429; lines 443-448).

Line 377. The sentence “Agricultural fields were not included in this study” is

repeated in line 381.

56. We have now removed the repeated statement.

Line 382. Change “nearby” to “near”

57. Changed.

Subsection Environmental factors (from line 403):

Lines 407-412. The description of the general methods for soil analysis should be moved to the following subsection (renamed as Soil analysis). Also, include a new heading for general soil analysis.

58. Thanks for the suggestion. We have now moved the soil analysis to a separate subsection with heading (lines 443-448).

The number of soil samples for each contaminant analysis is not clear, as different numbers are indicated in each subsection of the specific contaminant (see lines 422-423, 438-442, 462-463, 501 and 516). The number of soil samples for each contaminant should be clearly indicated in the subsection Field survey and soil sampling (lines 383-390). Avoid repetition in the respective subsection for each contaminant type.

59. Thanks for this important point. We have now clearly shown the number of soil samples for the analysis of each group of contaminants in the subsection of field survey and soil sampling, according to your and reviewer #1’s suggestions (lines 418-429; Table S2).

Line 425. Is “trace mixed acids” a mistake?

60. Sorry for the mistake, and it is “mixed acids” and we have revised the text accordingly.

Line 427. Change “using an” to “by”

61. Done.

Line 430. I think “aqua regia” should be written in italics.

62. We have now written it in italics.

Line 441. Change MP to microplastics

63. Thanks. It should be microplastics, but we have now removed this repeated sentence.

Lines 463-4. State what “5 mm” refers to: diameter, length of the longest side?

64. We have now clarified that “diameter” (line 485).

Line 468: change retain to maintain

65. Changed.

Line 475. Should it be “flotation” instead of “floatation”?

66. Used “flotation” instead of “floatation”.

Check for typing mistakes (see line 479).

67. We have corrected the typo. “The shape (i.e., fiber, fragment, and film) of the particles was assessed according to a previous method” (see lines 499-500).

Line 493. “carefully” seem to be in the wrong place.

68. We removed the “carefully”.

Lines 518-519. Remove sentence to avoid repetitions in the experimental procedure.

69. Done.

Line 546. Remove “)”

70. Done.

Line 550. Change “mean” to “indicate” or “represent” (“mean” is confusing here as it is also a statistical term).

71. Thanks for the suggestion. We have now changed it to “represent”.

Line 593. Remove the first “RMSEA”.

72. Done.

Figure 1. The sentence on lines 800-803 seems to be incorrectly written.

73. We have written that as “Changes (%) in soil contaminants in urban greenspaces compared with adjacent natural areas worldwide”.

Figure 2. This figure is not clear. What do the lines and arrows in blue and red mean in panel A? They are not yellow boxes, but orange. Some abbreviations are not specified.

74. Thanks. We have now consistently used black lines to avoid confusion, and also stated them as orange but not yellow boxes. We also specified all the abbreviations.

Thanks again for your help in improving our manuscript.

References

Bahram, M., Hildebrand, F., Forslund, S.K. et al. (2018) Structure and function of the global topsoil microbiome. *Nature* 560, 233–237

Guerra, C.A., Berdugo, M., Eldridge, D.J., Eisenhauer, N., Singh, B.K., Cui, H. et al., (2022) Global hotspots for soil nature conservation. *Nature* 610, 693-698.

Delgado-Baquerizo, M., Oliverio, A.M., Brewer, T.E., Benavent-González, A., Eldridge, D.J., Bardgett, et al., (2018) A global atlas of the dominant bacteria found in soil. *Science* 359, 320-325.

Delgado-Baquerizo, M., Eldridge, D.J., Liu, Y.-R., Sokoya, B., Wang, J.-T., Hu, H.-W., et al., (2021) Global homogenization of the structure and function in the soil microbiome of urban greenspaces. *Science Advances* 7, eabg5809.

Maestre, F.T., Le Bagousse-Pinguet, Y., Delgado-Baquerizo, M., Eldridge, D.J., Saiz, H., Berdugo, M., Gozalo, B., et al., (2022) Grazing and ecosystem service delivery in global drylands. *Science* 378, 915-920.

Mueller, P., Ladiges, N., Jack, A., Schmiedl, G., Kutzbach, L., Jensen, K., & Nolte, S. (2019). Assessing the long - term carbon - sequestration potential of the semi - natural salt marshes in the European Wadden Sea. *Ecosphere*, 10(1), e02556.

Howe, A. C., Jansson, J. K., Malfatti, S. A., Tringe, S. G., Tiedje, J. M., & Brown, C. T. (2014). Tackling soil diversity with the assembly of large, complex metagenomes. *Proceedings of the National Academy of Sciences*, 111(13), 4904-4909.

Li, G., Sun, G. X., Ren, Y., Luo, X. S. & Zhu, Y. G. Urban soil and human health: a review. *European Journal of Soil Science* 69, 196-215 (2018).

Tonn, B., Komainda, M. & Isselstein, J. (2021). Results from a biodiversity experiment fail to represent economic performance of semi-natural grasslands. *Nature Communications* 12, 2125

Reviewer #1 (Remarks to the Author):

The revised paper has been significantly improved as described in the rebuttal letter provided by the authors. The authors have addressed all points raised by the reviewers in a satisfactory manner. Nevertheless, I still find the scientific novelty somewhat limited as links between anthropogenic activities, land use and soil pollution are already well-studied at national, regional or continental scales. The survey would have had higher quality if a higher, and consistent, number of soil samples had been analyzed for all parameters. Finally, the many associations between different factors do not allow for inference of causal relationships. Hence, the implications of this work are somewhat vague although the findings may of course pave the way to the formulation of novel hypotheses for controlled studies as suggested by the authors.

Specific comments:

Line 177: rephrase "natural vs. anthropogenic" as the authors themselves agree (rebuttal letter) that this is not an appropriate phrase to use for this study.

Line 198: Rephrase "greater effect" as a causal relationship cannot be deduced from this association study.

Line 232: Spell out 2,4-D as this herbicide abbreviation may not be known by all readers.

Lines 378 and 382: Change "are" to "were".

Lines 383-384: This statement does not sound credible (sentence starting with "greenspace ...")
Figures are complex and a little difficult to read. Please prepare figures so they can stand alone without reference to text and insert units (especially Figure 1). "ARGS" should be changed to "ARGs" in Figure 1.

Responses to reviewers' comments

Reviewer #1 (Remarks to the Author):

The revised paper has been significantly improved as described in the rebuttal letter provided by the authors. The authors have addressed all points raised by the reviewers in a satisfactory manner. Nevertheless, I still find the scientific novelty somewhat limited as links between anthropogenic activities, land use and soil pollution are already well-studied at national, regional or continental scales. The survey would have had higher quality if a higher, and consistent, number of soil samples had been analyzed for all parameters. Finally, the many associations between different factors do not allow for inference of causal relationships. Hence, the implications of this work are somewhat vague although the findings may of course pave the way to the formulation of novel hypotheses for controlled studies as suggested by the authors.

1. We appreciated the reviewer highlighted that we have addressed all previous comments in a satisfactory manner. We also agree that future work including a larger number of samples will continue to move this research forward. To the best of our knowledge, however, this is the largest **paired global survey** aiming at comparing **multiple soil contaminants** in urban greenspaces vs. nearby reference “natural” areas, and this is different from many previous studies basing on unpaired sampling at a national/regional scale or concentrating on a few contaminants. Our work demonstrates that human-driven soil contamination in nearby natural areas mirrors that in urban greenspaces worldwide, and highlights that soil contaminants have the potential to cause dire consequences for ecosystem sustainability and human wellbeing. Although this study does not imply causality, it paves the way to the formulation of important hypotheses for controlled studies. This is all highlighted in the latest version of our manuscript.

Specific comments:

Line 177: rephrase "natural vs. anthropogenic" as the authors themselves agree (rebuttal letter) that this is not an appropriate phrase to use for this study.

2. Changed it to “geochemical background vs. anthropogenic” (line 177).

Line 198: Rephrase "greater effect" as a causal relationship cannot be deduced from this association study.

3. Changed it to “a greater contribution in explaining the proportion of...”(line 198).

Line 232: Spell out 2,4-D as this herbicide abbreviation may not be known by all readers.

4. Done: 2,4-Dichlorophenoxyacetic acid (lines 232).

Lines 378 and 382: Change "are" to "were".

5. Done (lines 378 and 382).

Lines 383-384: This statement does not sound credible (sentence starting with “greenspace ...” Figures are complex and a little difficult to read. Please prepare figures so they can stand alone without reference to text and insert units (especially Figure 1). “ARGS” should be changed to “ARGs” in Figure 1.

6. Thanks for the constructive comments. We have now modified the sentence as “Our findings indicate that soils in urban and natural ecosystems were similar in their levels of contamination” (lines 383). We did not make any claim about urban being similar to natural ecosystem for other soil variables not measured in this study. Finally, we also updated Figure 1 and checked other Figures for clarity without reference to the text.